



# TUNER-compliant error estimation for MIPAS

Thomas von Clarmann[1], Norbert Glatthor[1], Udo Grabowski[1], Bernd Funke[2], Michael Kiefer[1], Anne Kleinert[1], Gabriele P. Stiller[1], Andrea Linden[1], and Sylvia Kellmann[1]

[1]Karlsruhe Institute of Technology, Institute of Meteorology and Climate Research, Karlsruhe, Germany
[2]Instituto de Astrofísica de Andalucía, CSIC, Spain

**Correspondence:** Thomas von Clarmann (thomas.clarmann@kit.edu)

**Abstract.** This paper describes the error estimation for temperature and trace gas mixing ratios retrieved from the Michelson Interferometer for Passive Atmospheric Sounding (MIPAS) limb emission spectra. The following error sources are taken into account: measurement noise, propagated temperature and pointing noise, uncertainties of the abundances of spectrally interfering species, instrument line shape errors, and spectroscopic data uncertainties in terms of line intensities and broadening

coefficients. Furthermore, both the direct impact of volatile as well as persistent gain calibration uncertainties, offset calibration and spectral calibration uncertainties and their impact through propagated calibration-related temperature and pointing uncertainties are considered. An error source specific to the MIPAS upper atmospheric observation mode is the propagation of the smoothing error crosstalk of the combined NO and temperature retrieval. Whenever non-local thermodynamic equilibrium modelling is used in the retrieval, also related kinetic constants and mixing ratios of species involved in the modelling of populations of excitational states contribute to the error budget. Both generalized Gaussian error propagation and perturbation studies are used to estimate the error components. Error correlations are taken into account. Estimated uncertainties are provided for a multitude of atmospheric conditions. Some error sources were found to contribute both to the random and the systematic component of the total estimated error. The sequential nature of the MIPAS retrievals gives rise to entangled errors. These are caused by error sources that affect the uncertainty of the final data product via multiple pathways, i.e., on the one

hand directly, and on the other hand via errors caused in a preceding retrieval step. These errors tend to partly compensate each other. The hard-to-quantify effect of the horizontally non-homogeneous atmosphere and unknown error correlations of spectroscopic data are considered as the major limitations of the MIPAS error estimation.

## 1 Introduction

Availability of reliable and traceable uncertainty estimates is a precondition for quantitative scientific work with remotely

sensed atmospheric temperature and composition data. In order to serve this purpose, we present the scheme according to which error estimation is performed for the composition data retrieved from version 8 limb emission spectra recorded with the Michelson Interferometer for Passive Atmospheric Sounding (MIPAS, Fischer et al. 2008) on Envisat. This error estimation procedure refers to retrievals performed with the data processor developed and operated by the Institute of Meteorology and Climate Research (IMK) in cooperation with the Instituto de Astrofísica de Andalucía (IAA). A general description of the

processor is found in von Clarmann et al. (2003). The application to non-local thermodynamic equilibrium conditions is



documented in Funke et al. (2001, 2012). The processing of MIPAS spectra measured at reduced spectral resolution after an instrument failure in 2004 is described by von Clarmann et al. (2009). Kiefer et al. (2021) describe the first application to MIPAS version 8 spectra.

The error estimation of preceding MIPAS retrievals took into account all known major uncertainties but left room for im-
provement with respect to error correlation issues. It is the purpose of this paper to present a scheme that explores all available knowledge on the ingoing uncertainties, and, in particular, error correlations. We try to comply as much as possible with the recommendations on unified error reporting as specified by the SPARC Activity 'Towards Unified Error Reporting (TUNER)' presented in von Clarmann et al. (2020).

MIPAS temperature and tangent altitude pointing errors are already described in Kiefer et al. (2021). This paper provides
an update of their temperature error estimation scheme and aims at a general scheme for MIPAS trace gas retrieval error estimation. First the notation used is clarified (Section 2) and the error propagation schemes used for MIPAS are introduced (Section 3). After clarification of terminological issues (Section 4), we select the scenarios for which uncertainty estimates are carried out and how any arbitrary measurement is linked to these representative uncertainty estimates (Section 5). Then, for all relevant error sources, the respective error propagation scheme is discussed for the MIPAS temperature and tangent-altitude
pointing retrieval (Section 6) and retrieved mixing ratios of trace constituents (Section 7). The MIPAS ozone retrieval serves as an example to illustrate the application of the error estimation scheme (Section 9). In Section 10 we describe how representative error estimates are built from the sample of analyzed observations. The aggregation of component errors to total, random, and systematic error budgets is described in Section 11. Technical issues needed to make theory work are presented in Section 12. Finally, in Section 13 we summarize to which degree we succeeded in providing a robust error estimate and critically identify
issues that could not be solved in a satisfactory manner.

## 2 Definitions and notation

In agreement with the general concept by Rodgers (2000) and the notation suggested by von Clarmann et al. (2020) we use the following retrieval equation for MIPAS.

$$
\begin{aligned}
\hat{\boldsymbol{x}}_{i+1} \;=\; & \hat{\boldsymbol{x}}_i + \left(\mathbf{K}^T \mathbf{S}_{\mathrm{y,noise}}^{-1} \mathbf{K} + \mathbf{R}\right)^{-1} \\
& \left(\mathbf{K}^T \mathbf{S}_{\mathrm{y,noise}}^{-1}\left(\boldsymbol{y} - \boldsymbol{F}(\hat{\boldsymbol{x}}_i; \boldsymbol{b})\right) - \mathbf{R}(\hat{\boldsymbol{x}}_i - \boldsymbol{x}_{\mathrm{a}})\right)
\end{aligned}
\tag{1}
$$






where  $x$  is the vector of the target variable;

$\hat{x}$  is its estimate;

$i$  denotes the number of iteration;

$\mathbf{K}$  is the Jacobian with elements $\partial y_m / \partial x_n$;

$^T$  denotes transposed matrices;

$\mathbf{S}_{y,noise}$  is the measurement noise covariance matrix;

$\mathbf{R}$  is the regularization matrix;

$F$  is the radiative transfer function; in our case it is the KOPRA radiative transfer model (Stiller, 2000).

$b$  is a vector representing all other input parameters except the target variables of the retrieval;

and  $x_a$  is the vector representing the a priori knowledge on the target variables.

The inverse problem is decomposed species-wise. That is to say, after the retrieval of temperature and pointing information (Kiefer et al., 2021), ozone concentrations are retrieved in microwindows where the ozone signal is prominent and where interferences by other gases are low. As a next step, $H_2O$ concentrations are retrieved in spectral microwindows adequate for the $H_2O$ retrieval. In this manner the retrieval proceeds through the list of species, according to their dominance in the spectrum, where usually the concentrations of the pre-retrieved species are used for the retrieval of the gas currently under analysis. The latter we call 'target species'.

Within linear theory we have the averaging kernel matrix

$$\frac{\partial \hat{x}}{\partial x} = \mathbf{A} = \left( \mathbf{K}^T \mathbf{S}_{y,noise}^{-1} \mathbf{K} + \mathbf{R} \right)^{-1} \mathbf{K}^T \mathbf{S}_{y,noise}^{-1} \mathbf{K} \tag{2}$$

and the gain matrix

$$\frac{\partial \hat{x}}{\partial y} = \mathbf{G} = \left( \mathbf{K}^T \mathbf{S}_{y,noise}^{-1} \mathbf{K} + \mathbf{R} \right)^{-1} \mathbf{K}^T \mathbf{S}_{y,noise}^{-1} \tag{3}$$

For errors, covariances, etc., we use the TUNER notation: the first subscript denotes the quantity to which the error refers, and the second subscript denotes the source of the error. For example, $\mathbf{S}_{O3;noise}$ is the covariance matrix characterizing the ozone error component due to measurement noise. Beyond this, we use $\sigma_q$ for standard deviations of a generic variable $q$, and $\Delta q$ for perturbations with a sign, where

$$|\Delta_b q| = \sigma_{q;b} \tag{4}$$





and where $b$ denotes the errors source.

## 3 Error propagation

MIPAS retrievals depend on measured spectra and auxiliary information which both are uncertain. Depending on the uncertainty information available, different schemes to estimate error propagation are in use. All error estimations used for MIPAS rely on linear error estimation.

Gaussian error propagation is applied to measurement errors of which the complete covariance information in the measurement domain is available:

$$\mathbf{S}_{\text{x;meas}} = \mathbf{G}\mathbf{S}_{\text{y;meas}}\mathbf{G}^T, \tag{5}$$

where $\mathbf{S}_{\text{x;meas}}$ is the covariance matrix representing the error of the retrieved variables caused by the measurement error $\mathbf{S}_{\text{y;meas}}$. In this formulation $\mathbf{S}_{\text{y;meas}}$ is a placeholder and will be replaced by the specific type of measurement error under assessment.

The gain matrix $\mathbf{G}$ does not only include the sensitivities of the target gas with respect to the measurement but also the sensitivities of all variables which are fitted along with the target gas in the same inversion, e.g., background continua or further gases that are simultaneously fitted. These gain matrices are available from the retrieval and do not have to be newly evaluated. The resulting covariance matrix $\mathbf{S}_{\text{x;meas}}$ includes also entries for all joint-fit variables that are fitted along with the target variables for various reasons.

For parameter errors whose covariance information in the altitude domain is available, the parameter uncertainties can be linearly mapped into the measurement domain, and then propagated into the target variable space:

$$\mathbf{S}_{\text{x;b}} = \mathbf{G}\mathbf{K}_{\text{b}}\mathbf{S}_{\text{b;meas}}\mathbf{K}_{\text{b}}^T\mathbf{G}^T, \tag{6}$$

where $\mathbf{S}_{\text{x;b}}$ is the covariance matrix representing the error in the retrieved variables caused by the parameter error $\mathbf{S}_{\text{b}}$; $\mathbf{K}_{\text{b}}$ is the Jacobian, representing the sensitivities of the radiance in the analysis windows of the target gas $\boldsymbol{x}$ with respect to changes of the parameter profile $\boldsymbol{b}$. If $\boldsymbol{b}$ is obtained in a preceding retrieval of the sequential retrieval chain, it has to be noted that $\mathbf{K}_{\text{b}}$ is not the same as the Jacobian used in the preceding retrieval of $\boldsymbol{b}$ because it refers to the spectral radiances in the microwindows used for the retrieval of the target gas $\boldsymbol{x}$, as opposed to those used in preceding retrieval of $\boldsymbol{b}$. Thus, it is not available as a by-product of the temperature and pointing retrieval but needs to be calculated with a dedicated call of the forward model.

Sensitivity studies based on the difference between perturbed spectra ($\boldsymbol{F}_{\text{perturbed}}$) and nominal spectra ($\boldsymbol{F}_{\text{nominal}}$) are an alternative when the methods presented above are not applicable or inadequate. Within linear error estimation, the estimated error is proportional to the spectral difference caused by the erroneous quantity. The effect of a 1sigma uncertainty of any parameter $b_k$ affecting the signal $y$ can be estimated by perturbing $b_k$ in the input of a run of the forward model $\boldsymbol{F}(\boldsymbol{x}, \boldsymbol{b})$ by the respective $\Delta\boldsymbol{b}$. The response of the retrieval to this perturbation is then estimated as

$$\Delta_{\text{b}_k}\boldsymbol{x} = -\mathbf{G} * (\boldsymbol{F}_{\text{perturbed}} - \boldsymbol{F}_{\text{nominal}}). \tag{7}$$





Here $F_{\text{nominal}}$ are the radiances simulated with the input data and results of the retrieval of $x$, while $F_{\text{perturbed}}$ are the spectral radiances obtained after perturbation of the parameter(s) under assessment by $1\sigma$. Many applications of Eq. (7) are sign-sensitive. That is to say, contrary to the application of variances, the signs of the elements of $\Delta_{b_k} x$ have to be considered. The

negative sign on the right-hand side of Eq. 7 is due to the fact that perturbations are applied to the $F(\hat{x}_i; b)$ term in Eq. 1, which appears with a negative sign there. In order to avoid confusion with respect to signs, we consistently use the convention that instrumental uncertainties such as gain calibration or instrument line shape uncertainties are understood as uncertainties of the related model parameters and thus refer to the $F(\hat{x}_i; b)$ rather than the $y$ term in Eq. 1. The sign of the perturbation as such is arbitrary. Only self-consistence is of concern when a perturbation enters the error estimation in multiple pathways as

discussed in Section 4.4.

Also here, the gain matrix $\mathbf{G}$ of the retrieval has to include not only entries considering the target gas $x$ but also those related to all joint-fit variables.

For species whose mixing ratios are retrieved in the logarithmic domain ($H_2O$, CO, NO, $NO_2$; in middle atmosphere (MA), upper atmosphere (UA), and noctilucent cloud (NLC) measurement modes[1] also $O_3$; for specific MA research products also

$CH_4$ and $N_2O$) also the error estimates are performed in the logarithmic domain and finally mapped into the mixing ratio domain.

## 4   Terminology

Since no compelling argument has been provided that 'uncertainties' and 'estimated errors' connote different concepts (von Clarmann et al., 2022), we use these terms almost as synonyms. The only subtle linguistic difference seems to be that 'uncer-

tainty' is an attribute of the measurand, i.e., the atmospheric state, which is not known with certainty, while the 'error' is an attribute of the measurement. Under normal conditions, we know the measurement with certainty, but the measurement is not perfectly correct. The error of the measurement causes an uncertainty in our knowledge of the atmospheric state. We kept our terminology in broad agreement with common language since Gauss (1809), although this is in conflict with the stipulation by the Joint Committee for Guides in Metrology (JCGM) (2008).

In agreement with the definitions suggested by TUNER (von Clarmann et al., 2020) we distinguish random errors and systematic errors. Beyond these, we also have to deal with the so-called headache errors and entangled errors. These concepts will be introduced in the following. For reasons discussed in von Clarmann (2014), we do not include the smoothing error in our error budget.

### 4.1   Random errors

Random errors are errors that cause a standard deviation of the differences between independent coincident measurements. Thus, the random error budget exceeds measurement noise and includes randomly varying parameter errors. Chief contributors are: measurement noise, tangent altitude uncertainties, the volatile component of gain calibration uncertainty, offset calibration

---

[1]Definitions and details of the measurement modes are compiled in Oelhaf (2008).





uncertainty, spectral shift uncertainty and uncertainties in the abundances of gases that are kept fixed in the retrieval of the target gas.

## 4.2 Systematic errors


According to TUNER terminology, systematic errors are those errors that cause, in the long run, a bias between independent measurements of the same state variable at the same time and place. Error correlations in the altitude domain, or between different error components related to different error sources are irrelevant for the classification as systematic error. Even correlations in the time domain on a short time-scale do not make an error systematic as long as it does not cause a bias in the long

run.

The advantage of this definition is that, contrary to other definitions of this term, the systematic error is observationally significant in the sense that it is accessible by observations and thus empirically testable. Other definitions of systematic errors run risk to lead to theoretical quantities that are recalcitrant against empirical testing. Chief contributors to the MIPAS systematic error budget are uncertainties in spectroscopic data with respect to the intensities and broadening coefficients of the

spectral lines used, uncertainties in the MIPAS modulation efficiency that leads to uncertainties in the instrument line shape, and the persistent part of the gain calibration uncertainty, which is dominated by detector nonlinearity issues (Kleinert et al., 2018, their Table 3).

## 4.3 Headache errors

Arguably, the distinction between random and systematic errors is not always quite clear, because, e.g., nonlinear propagation

of random errors can cause a bias, and a random modulation of a systematic error can cause some scatter. Since related difficulties can cause some headache, we call these errors 'headache errors'.

We assume that MIPAS retrievals are moderately nonlinear and thus linear theory is sufficient for error estimation. Within this framework, any possible bias due to the nonlinearity of the retrieval – we call this 'type A headache error' – is inaccessible.

Conversely, the retrieval may depend in a deterministic way on some uncertain input quantity, which is the same for all

retrievals. Ideally, this would give rise to a systematic error in the estimate. In the real world, however, the impact of the uncertain input quantity can depend on further quantities which may vary randomly. This causes a random modulation of the initially systematic error. The resulting error causes both a bias and a standard deviation of differences. We call this type of error 'type B headache error'. We assess this type of error using statistics over a sample of test cases. An example of a type B headache error is an initially systematic error due to the spectroscopic data of an interfering species of randomly varying

concentration[2].

In order to avoid propagating the related headache to the data users, the systematic and random components of the headache errors will be listed separately in the systematic and random error budgets.

---

[2]The authors are grateful to N. J. Livesey, who has mentioned this example in a discussion.


## 4.4 Entangled errors

Some errors in the input parameters enter the error budget of the target quantity via multiple pathways. This is because the
MIPAS retrievals are performed sequentially. In the first step, temperature and pointing information are retrieved; in the second
step this information is used for the retrieval of ozone distributions. The retrieved temperature, pointing, and ozone information
is used for the subsequent retrievals of the concentrations of other gases. Errors have to be propagated through this retrieval
chain. Some uncertainties affect the target retrieval directly, as well as indirectly, because it has already affected a quantity
retrieved earlier in the retrieval chain. For example, the gain calibration uncertainty has a direct effect on the retrieved target
gas. A positive perturbation, representing an overestimation of the radiative gain in the radiative transfer forward modelling,
makes the term $F_{\mathrm{perturbed}}$ in Eq. (7) larger, resulting typically in a negative error $\Delta_{\mathrm{gain}}x$ of the target species concentration. A
positive gain perturbation, however, entails also a smaller retrieved temperature, and the too small temperature requires a larger
amount of the target gas to fit the measured target lines. Formally speaking, the negative temperature perturbation implies
a positive error component $\Delta_{\mathrm{T,gain}}x$ of the target gas due to the propagated gain-induced temperature error. Thus, the direct
gain error and the propagated gain-induced temperature error mutually counteract and tend to compensate each other. We call
these error components which enter the error budget via multiple pathways 'entangled errors'. They must not be treated as
independent errors because then related error compensation information would be lost.

We distinguish between two kinds of entangled errors. The first variant of the entangled errors affects only spectral signatures
of the species associated with this error. This is best illustrated by use of an example. The line intensities, as part of the
spectroscopic data used, of a pre-fitted gas may be too low. This typically results in too high concentrations in the retrieval of
this gas. In the microwindows used for the target gas analysis these erroneous line intensities affect only the signal caused by
the pre-fitted species. Within linear theory and with favorable assumptions on the correlations of line intensity errors of the
pre-fitted gas in force (same relative error of line intensities in the whole spectral range), the direct error and the propagated
error through the use of the pre-fitted concentration cancel out almost perfectly. We call this variant of entangled errors 'lazy
entangled errors'. Due to its cancellation characteristics, we do not consider these in the MIPAS error budget, except where
explicitly mentioned otherwise.

The other variant of the entangled errors affects all radiances used in the target gas retrieval, regardless to which gas the
spectral signatures belong. Again we use an example to illustrate this mechanism. Too low assumed $CO_2$ concentrations in
the temperature retrieval may cause too high retrieved temperatures. The use of these too high temperatures in the target gas
retrieval affects all radiances used in the target gas retrieval, not only the signatures associated with $CO_2$. The compensation
mechanism discussed above is confined to those parts of the signal used for the target gas retrieval where $CO_2$ has some
signal interfering with the target signal, and no full cancellation of the temperature error component induced by erroneous $CO_2$
mixing ratios takes place. We call this variant of entangled errors 'serious entangled errors'. Since these are the only entangled
errors considered in the MIPAS error budget, the term 'entangled error' always refers to a serious entangled error.
To account for the entangled nature of these errors, their impact is estimated using one single perturbation spectrum per
tangent altitude, where both the direct and the propagated error terms are included.





## 5 Selected reference scenarios

Due to operational constraints it is not possible to provide full error estimates for each single measurement. Instead, the errors are evaluated for representative classes of cases. These include all relevant combinations of latitude band, season and illu-
mination, and, where relevant, solar activity. Since it is questionable if a single limb scan can safely represent a large class of measurements, we consider multiple limb sequences for each scenario, and the average errors are used as representative error estimates. For polar and midlatitudinal conditions the following seasons were considered: northern spring / austral autumn (Mar, April, May), northern summer / austral winter (June, July, August), northern autumn / austral spring (September, October, November), and northern winter /austral summer (December, January, February). For tropical conditions no dis-
tinction according to the season is made. Further, we distinguish between daytime and nighttime situations. Neither twilight conditions nor latitudes that could not be clearly assigned to a typical scenario were considered. Tables A1–A5 present the orbits of which reference limb scans were chosen for error estimation, as well as latitude ranges and ranges of solar zenith angles (SZA) selected for error estimation. Table A1 refers to high resolution (HR, employed from 2002–2004) nominal mea-surements, Table A2 to reduced resolution (RR, employed from 2005–2012) nominal measurements, Table A3 to RR middle
atmosphere measurements, and Tables A4 and A5 to RR upper atmosphere measurements under low and high solar activity. For the upper-troposphere / lower-stratosphere measurement mode no dedicated error analysis has been made, because the RR-nominal mode error estimates are deemed well representative also for this particular observation mode. Similarly, MA error estimates are deemed well representative for NLC mode measurements. The number of orbits selected was chosen such that for each scenario at least 33 limb scans with converged retrievals and a low lowermost valid tangent altitude were available.

Each MIPAS measurement has been assigned to a reference class (scenario), for which representative error estimates are available, following the criteria defined in Tab. A6, in terms of season, latitude, and solar zenith angle. Since all MIPAS measurements have to be assigned to a representative case, the respective classes are larger in terms of latitudinal coverage and solar zenith angle coverage than those used to evaluate the errors.

  Based on this classification, systematic and random error budgets are estimated for each single measurement. For errors of a
multiplicative nature the respective error estimates are obtained by scaling the relative error with the actual constituent profile retrieved for the geolocation under assessment.

## 6 Error components of the temperature and pointing retrieval

Temperature is the first atmospheric state variable that is retrieved in the sequential retrieval chain. Temperature is retrieved along with line-of-sight pointing information in terms of tangent altitudes from $CO_2$ emissions. Since temperature and pointing
information are retrieved in one step, we represent this information in one single vector $\boldsymbol{TLOS}$. The retrieval technique and error estimation is documented in Kiefer et al. (2021). Therefore, for most error sources a cursory discussion must suffice here. Updates to the temperature and pointing error estimation refer to the uncertainties of the abundances of interfering trace gases, the treatment of the gain calibration uncertainty, the uncertainty of zero-level calibration in terms of an additive radiance offset, and, for measurement modes involving explicit non-local thermodynamic equilibrium (non-LTE) modelling, uncertainties of



related specific parameters, particularly rate constants of the kinetic processes and concentrations of trace gases that govern non-LTE related processes.

## 6.1 Measurement noise in temperature retrievals

The noise error covariance matrix $\mathbf{S}_{\mathrm{TLOS;noise}}$ of the target gas profile $\boldsymbol{TLOS}$ is calculated from the gain matrix of the retrieval, $\mathbf{G}_{\mathrm{TLOS}}$, and the measurement noise covariance matrix, $\mathbf{S}_{\mathrm{y;noise}}$, using generalized Gaussian error propagation (Eq. 5). Technically speaking, we use for MIPAS error estimation the following variant of this equation

$$\mathbf{S}_{\mathrm{TLOS;noise}} = \mathbf{A}_{\mathrm{TLOS}}(\mathbf{K}_{\mathrm{TLOS}}^{T}\mathbf{S}_{\mathrm{y;noise}}\mathbf{K}_{\mathrm{TLOS}} + \mathbf{R}_{\mathrm{TLOS}})^{-1}, \tag{8}$$

which is fully equivalent to the application of Eq. (5) to $\mathbf{S}_{\mathrm{y;noise}}$ but technically more efficient, because it relies on quantities that were stored during the retrieval and thus have not to be calculated again during the error estimation procedure. Here subscript TLOS indicates that the respective matrices refer to the retrieval of $\boldsymbol{TLOS}$. Obviously, $\mathbf{S}_{\mathrm{y;noise}}$ here and henceforth is understood to contain only variances and covariances referring to spectral gridpoints actually included in the spectral microwindows used in the retrieval under assessment. Formally, the retrieval vector, gain function, and resulting covariance matrix contain also entries referring to some further fit variables (see, Kiefer et al., 2021 for details), but finally only the block representing the retrieved temperature profile and the retrieved tangent altitudes, as well as covariances between these quantities are relevant.

The measurement noise covariance matrix $\mathbf{S}_{\mathrm{y;noise}}$ characterizes the noise in all spectral grid-points used for the retrieval of trace gas profile $\boldsymbol{x}$ at all tangent altitudes of the limb sequence used. Information on measurement noise is provided by ESA along with the measured spectra. It has been estimated from the high-pass filtered imaginary part of the complex calibrated spectra.

Originally, measurement noise is independent between all data points, entailing a diagonal $\mathbf{S}_{\mathrm{y;noise}}$ matrix. However, since the IMK/IAA processor uses spectra apodized with the Norton and Beer (1976) 'strong' apodization function, the measurement noise covariance matrix $\mathbf{S}_{\mathrm{y;noise}}$ is manipulated accordingly:

$$\mathbf{S}_{\mathrm{y;noise}} = \mathbf{Q}\mathbf{S}_{\mathrm{y;noise;\ unapodized}}\mathbf{Q}^{T}, \tag{9}$$

where $\mathbf{Q}$ is the rotationally symmetric matrix representing the discrete convolution with the apodization function.

Measurement noise contributes to the random error. Due to the structure of the gain matrix $\mathbf{G}_{\mathrm{TLOS}}$, $\mathbf{S}_{\mathrm{TLOS;noise}}$ typically has significant non-zero off-diagonal entries characterizing error correlations in the altitude domain. Even for a diagonal $\mathbf{S}_{\mathrm{y;noise}}$ matrix, these entries would not disappear, because the limb sounding geometry implies a $\mathbf{G}$ without diagonal structure. For the non-diagonal $\mathbf{S}_{\mathrm{y;noise}}$ matrix of apodized spectra, the non-diagonality of $\mathbf{S}_{\mathrm{TLOS;noise}}$ holds *a fortiori*.

## 6.2 Uncertainties of interfering species in the temperature retrieval

The spectral microwindows used for the MIPAS temperature and tangent altitude retrieval are dominated by $CO_2$ lines but contain some signal of other gases. Both for $CO_2$ and for the interfering gases, the temperature and tangent altitude retrieval has to rely on assumptions. Related uncertainties propagate onto the retrieved temperatures and tangent altitudes. We have different





sources of information on these trace gas abundances. These are model calculations for $CO_2$, older versions of MIPAS retrievals for gases included in the MIPAS data product, and the MIPAS first guess database, for gases not included in the MIPAS data product but still contributing as interferent.

### 6.2.1   $CO_2$ information from model runs

For temperature and tangent altitude retrievals, we use $CO_2$ mixing ratios calculated with the Whole Atmosphere Community Climate Model (WACCM, Marsh 2011; Marsh et al. 2013) version 4, run for specified dynamics (García et al., 2017) and uncertainties as reported by Kiefer et al. (2021). Related $\boldsymbol{TLOS}$ uncertainties are estimated using Eq. (7) with

$$\boldsymbol{F}_{\text{perturbed}} = \boldsymbol{F}(\boldsymbol{TLOS}; \boldsymbol{b}_{\text{CO2}} + \Delta\boldsymbol{b}_{\text{CO2}}), \tag{10}$$

where $\boldsymbol{b}_{\text{CO2}}$ and $\Delta\boldsymbol{b}_{\text{CO2}}$ are the $CO_2$ profiles used and their 1sigma perturbations. The perturbation is performed in one step,
covering all altitudes. The mixing ratio of the interfering species under assessment is perturbed at all altitudes by 1 $\sigma$ of its uncertainty at this altitude. As a conservative estimate, all these perturbations are applied with the same sign. This is admittedly not ideal but we have no more specific correlation information available that would allow for a more adequate approach. Our treatment usually provides an upper estimate of the propagated errors.

For MIPAS measurements of the nominal and UTLS-1 measurement modes, $CO_2$ mixing ratio uncertainties are deemed
to contribute to the random error, because below about 70 km this error component is thought to be chiefly caused by the natural variability around the climatological value. At altitudes above about 70 km altitude, $CO_2$ mixing ratio uncertainties are supposed to be dominated by a pronounced systematic component due to model biases, which has to be considered when spectra recorded in the middle and upper atmosphere measurement modes are analyzed.

### 6.2.2   Abundance information from previous data versions in temperature retrievals

The MIPAS version 5 data product, the predecessor of the current version V8, includes abundance and uncertainty information on many species contributing to the infrared spectrum. The respective abundance information of the limb scan under assessment is used for the interfering species in the forward calculations of the version V8 retrievals. Although broadly considered to be of inferior quality compared to V8 data, the V5 data are considered good enough to be used to characterize the small contributions of the interfering species to the signal in the microwindows of the target gas retrieval (Kiefer et al., 2021). While originally the
mapping of the uncertainties of the interfering species was estimated using perturbation calculations (Eq. (7), we now use the covariance information $\mathbf{S}_{\text{b;noise}}$ that is available from the preceding data retrievals as $\mathbf{S}_{\text{b;meas}}$ in Eq. (6). The error components due to uncertainties in the concentrations of interfering species from version 5 MIPAS retrievals contribute chiefly to the random error of the target gas retrieval.

### 6.2.3   Abundance information from the initial guess data base in temperature retrievals

For interfering gases except $CO_2$ that are not available from preceding MIPAS data versions, the retrievals use mixing ratios from the initial guess database (Kiefer et al., 2002, and updates thereof). Available uncertainty information is vague, and often


"educated guesses" or "rational agents' personal beliefs" have to be used. Typically, no correlation information is available. Due to this lack of correlation information in the altitude domain, error estimation is approximated by a perturbation calculation using Eq. (7), as described in Section 6.2.1.

The capability of the radiative transfer model KOPRA to provide Jacobians with respect to gas concentrations helps to avoid a separate radiative transfer calculation with perturbed concentration data for each trace gas. Instead, the partial derivatives $\frac{\partial y_i}{\partial vmr_{j;g}}$ of spectral radiances $y_i$ with respect to the volume mixing ratio of gas $g$ at altitude $j$ are extracted and used for a linear approximation of the perturbation spectrum $\boldsymbol{F}_{\text{perturbed},g}$ as

$$\boldsymbol{F}_{\text{perturbed},g} = \boldsymbol{F}_{\text{nominal}} + \sum_j \Delta vmr_g, j \frac{\partial \boldsymbol{y}}{\partial vmr_{j;g}} \tag{11}$$

In most cases the resulting error components contribute less than one percent to the total error budget and therefore are deemed negligible. Since the true errors of this category are most likely even smaller than our estimates, this holds *a fortiori*.

For the category of interferents discussed here, no vertical correlation information is available, and everything said in the context of $CO_2$ uncertainties applies *mutatis mutandis*.

We assume that the error in the retrieved quantities caused by using concentrations from a database is dominated by natural

variability. That is to say, we assume that the database provides, on average, in the long run, the correct values, and that the related error is driven by the difference between the actual state and the mean state. With this supposition in force, related errors contribute to the random error budget. Admittedly, this supposition can be challenged, but the contribution of this category of interferents to the total error budget is so small that a more detailed assessment does not seem justified.

### 6.3  Calibration uncertainties in temperature retrievals

Under calibration uncertainties we summarize gain calibration uncertainties, zero offset calibration uncertainties, frequency shift uncertainties and instrument line shape uncertainties.

### 6.3.1  Gain calibration uncertainties in temperature retrievals

MIPAS gain calibration relies on reference measurements involving an internal blackbody of known temperature and deep space measurements. Related uncertainties have random and systematic components. The random component includes noise

in the blackbody measurements and gain variation between blackbody measurements. The systematic component includes inaccuracies of the calibration blackbody, the errors in the correction of the detector nonlinearity, and the neglect of higher order artefacts (Kleinert et al., 2018). Table 3 of their paper allows to calculate the random and systematic uncertainties separately. Estimated random uncertainties are 0.2%, 0.2%, 0.2%, 0.2% and 0.4% for the MIPAS A band (685-980 cm$^{-1}$), AB band (1010-1180 cm$^{-1}$), B band (1205-1510 cm$^{-1}$), C band (1560-1760 cm$^{-1}$), and D band (1810-2410 cm$^{-1}$), respectively.

The corresponding systematic uncertainties are 1.1%, 1.0%, 1.0%, 0.3% and 0.3%, respectively. Apparent discrepancies of the values are explained by the fact that the values reported by Kleinert et al. (2018) are to be understood as 2sigma uncertainties, while our error estimation is consistently based on 1sigma uncertainties.





Contrary to the approach described in Kiefer et al. (2021), the gain calibration error $\Delta_{\mathrm{gain}}\boldsymbol{TLOS}$, of the temperature and tangent altitude vector $\boldsymbol{TLOS}$, is now estimated separately for its random and its systematic component by application of Equation (7). The perturbations are the same for all spectra of the limb sequence under assessment. We get

$$
\begin{aligned}
\Delta_{\mathrm{gain}}\boldsymbol{TLOS} &= -\mathbf{G}(\boldsymbol{F}_{\mathrm{perturbed}} - \boldsymbol{F}_{\mathrm{nominal}}) \\
&= -\mathbf{G}\left(\left(1 + \left(\frac{\Delta y}{y}\right)_A\right)\boldsymbol{F}_{\mathrm{nominal}} - \boldsymbol{F}_{\mathrm{nominal}}\right),
\end{aligned}
\tag{12}
$$

where $\boldsymbol{F}_{\mathrm{perturbed}}$ are the spectral radiances used for the retrieval, of all involved tangent altitudes, with the gain perturbation applied. The gain uncertainty of the MIPAS A band, which is used for the temperature and tangent altitude retrieval, is represented by the scalar $\left(\frac{\Delta y}{y}\right)_A$. $\boldsymbol{F}_{\mathrm{nominal}}$ are the radiances calculated witht the radiative transfer model KOPRA for the actual limb sequence under assessment.

### 6.3.2 Zero offset calibration uncertainties in temperature retrievals

On the face of it, it may seem inadequate to consider the zero offset calibration uncertainty in the error budget, because a zero offset correction is jointly retrieved along with the target variables (Kiefer et al., 2021). We have, however, to distinguish between two different mechanisms causing offset uncertainty, namely the approximately wavenumber-independent component and the offset noise.

The approximately wavenumber-independent component of the offset uncertainty is caused by a possible offset drift between calibration measurements. This error component is indeed accounted for by the retrieval of the offset correction along with the retrieval of the target quantities, because this additive offset correction assumes wavenumber-independence within each microwindow (von Clarmann et al., 2003). Thus, this error component does not need to be considered in the error budget.

The situation is different for the noise in the deep space measurements that are used for the zero-offset calibration. This noise has a spectral dependence and is thus not fully corrected for by the offset retrieval mentioned above. It is for this reason that we have updated the error estimation scheme for temperature and tangent altitude retrievals since (Kiefer et al., 2021) to include also the noise component of the offset calibration. It is estimated using Eq. (5), where $\mathbf{S}_{\mathrm{TLOS;meas}}$ is specified as

$$
[\mathbf{S}_{\mathrm{y;offset}}]_{i;j} = nesr_i * nesr_j * r_{i;j},
\tag{13}
$$

where $nesr_i$ and $nesr_j$ are the noise equivalent spectral radiances of the offset measurements at spectral gridpoints $i$ and $j$ (see Kleinert et al., 2018). Error correlations occur due to apodization, due to the fact that offset calibration measurements are provided at a shorter maximum optical path difference than the scene measurements, and because the same offset measurement is used for multiple tangent altitudes. To calculate the respective correlation coefficients $r_{i;j}$, first the apodization function is convolved with $\sin(i\pi*c)/(i\pi*c)$, where $i$ is the index of the spectral gridpoint, and where $c$ is the contrast between maximum optical path differences of the deep space and the scene measurements. For MIPAS high resolution, reduced resolution bands A-C and reduced resolution band D we have $c_{HR} = 0.1$, $c_{RR(AC)} = 0.2$, and $c_{RR(D)} = 0.033$, respectively. This convolution product is convolved with itself and then normalized such that its maximum is 1. The $k$th value beside the maximum is used as





$r_{i,i\pm k}$. For correlations in the altitude domain, it has to be considered that measurements recorded during a forward movement of the interferometer mirror, are all offset-calibrated with a deep space measurement with forward movement of the mirror, and backward scene measurements are calibrated with a backward deep space measurement. This implies that measurements at every second tangent altitude rely on the same deep-space spectrum for offset calibration. In consequence, the entry in $[\mathbf{S}_{y;\text{offset}}]_{i;j}$ is zero when the indices point at an odd and an even tangent altitude but does not depend on the tangent altitude as long as $i$ and $j$ both point at even or both at odd tangent altitudes. That is to say, the correlation coefficient $r_{i;j}$ between data points of two odd or two even tangent altitudes is the same for all pairings of the same spectral distance, regardless if $i$ and $j$ belong to the same tangent altitude or not.

Although the same offset calibration measurement is used for a couple of scene spectra, causing some error correlations in the time domain, in the long run the offset error contributes to the standard deviation of the differences between to independent measurement systems. Therefore, it is regarded as random error.

### 6.3.3 Spectral shift uncertainties in temperature retrievals

Prior to the retrievals, a correction of the frequency calibration of the MIPAS spectra is performed for each limb scan, using narrow, isolated lines spread over the entire spectrum covered by MIPAS. A linear model is fitted to the individual spectral shifts determined for each of these lines, providing a linear relation between the spectral shift and wavenumber. The scatter of the individual spectral shifts around the regression line serves as an estimate of the residual frequency correction uncertainty and was found to be 0.00029 cm$^{-1}$ (Kiefer et al., 2021). The resulting error in the mixing ratio of the target gas is evaluated using Eq. 7, where $\boldsymbol{F}_{\text{perturbed}}$ is an estimate of the spectral radiances for a spectral shift perturbed by the 1sigma frequency correction uncertainty, and where $\boldsymbol{F}_{\text{nominal}}$ are the spectral radiances calculated with the nominal frequency correction.

Errors in the retrieved quantities due to the spectral shift uncertainty contribute to the random error. They are fully correlated in the altitude domain.

### 6.3.4 Instrument line shape uncertainties in temperature retrievals

Errors in the line shape used in the radiative transfer calculation are quantified in terms of modulation efficiency, which is a key input parameter of the instrument line shape model used (Hase, 2003). The propagation of the modulation efficiency error is estimated using Eq. (7), where

$$\boldsymbol{F}_{\text{perturbed}} = \boldsymbol{F}(\boldsymbol{TLOS}; e + \Delta e) \tag{14}$$

and where

$$\boldsymbol{F}_{\text{nominal}} = \boldsymbol{F}(\boldsymbol{TLOS}; e). \tag{15}$$

$\boldsymbol{TLOS}$ Scalar $e$ is the nominal modulation efficiency and $\Delta e$ its perturbation by 1 $\sigma$.

The component of the instrument line shape error related to the phase does not need to be considered explicitly, because it affects the frequency shift only and thus is implicitly included in $\Delta_{shift}\boldsymbol{x}$.





Since the modulation efficiency parameter is based on a pre-flight study and used for all MIPAS retrievals, the related error in the target mixing ratio profile is systematic and fully correlated in the altitude domain.

### 6.4 Uncertainties in spectroscopic data in temperature retrievals

The leading components of uncertainties in spectroscopic data are line intensity uncertainties and uncertainties in the broadening coefficients. Both error sources are evaluated independently. The fact that no correlation information on spectroscopic
uncertainties is available is a major drawback. For most retrievals from MIPAS data multiple lines are used. If the intensity errors of multiple lines were uncorrelated, e.g., because they are dominated by measurement noise in the lab measurements, then their effect would partly average out in a multi-line retrieval. Conversely, if the intensity errors were strongly correlated, e.g., because they are caused by uncertainties of the amount of gas in the cell in the laboratory measurement, then their effect would be systematic and thus survive the implicit averaging taking place in a multi-line retrieval. Similar considerations hold for the
broadening coefficients. Since we have no better information, we consider the spectroscopic uncertainties as fully correlated. This assumption is conservative with respect to the error budget of the target but optimistic as far as error compensation in the context of entangled errors may be over-estimated. Since the same spectroscopic parameters are used for all MIPAS retrievals of a certain target gas, related concentration errors contribute chiefly to the systematic error budget. They are fully correlated in the altitude domain, as far as the same microwindows are used for all altitudes.

#### 6.4.1 Line intensity uncertainties in temperature retrievals

The response of the retrieval of temperature and tangent altitudes to errors of $CO_2$ line intensities is estimated by perturbation using Eq. (7). $\boldsymbol{F}_{\mathrm{perturbed}}$ are the spectral radiances calculated with the intensities of all $CO_2$ lines of the target gas all perturbed by $1\,\sigma$ of their individual uncertainty, where all perturbations have the same sign. $\boldsymbol{F}_{\mathrm{nominal}}$ are the spectral radiances calculated with the nominal line intensities.

#### 6.4.2 Broadening coefficient uncertainties in temperature retrievals

All said for the propagation of line intensity uncertainties holds *mutatis mutandis* also for the uncertainties of broadening coefficients. For the evaluation of the error component due to the target gas broadening coefficients according to Eq. (7), we calculate $\boldsymbol{F}_{\mathrm{perturbed}}$ with the broadening coefficients of the target gas all perturbed by $1\,\sigma$ of their individual uncertainty, where all perturbations have the same sign.

## 7 Error components of the trace constituents retrieval

For MIPAS trace gas retrievals, the following error sources are considered: Measurement noise; uncertainties in temperature and pointing information ($\boldsymbol{TLOS}$); uncertainties in the mixing ratios of interfering species which contribute in a sizeable way to the signal in the analysis window(s) of the target species and are not jointly fitted with the target gas; under certain conditions also smoothing error crosstalk can be an issue. Further error sources under consideration are gain, offset, and





frequency calibration errors as well as instrument line shape uncertainties; uncertainties in spectroscopic data in terms of line intensity and broadening coefficients; and, if applicable, uncertainties of specific parameters to non-local thermodynamic equilibrium (non-LTE), such as kinetic rate constants or abundances of trace gases that interact via specific non-LTE processes rather than by spectral interference. In the following sections, the related error propagation schemes are discussed.

### 7.1  Measurement noise in the trace constituents retrieval

Noise in retrieved trace gas abundance $x$ is calculated by Gaussian error propagation, using the same method as discussed for the temperature retrieval in Section 6.1. The noise error covariance matrix $\mathbf{S}_{\mathrm{x;noise}}$ of the target gas profile $x$ is calculated from the gain matrix of the retrieval, $\mathbf{G}$, and the measurement noise covariance matrix, $\mathbf{S}_{\mathrm{y;noise}}$, using generalized Gaussian error propagation (Eq. 5). Again, we use for MIPAS error estimation the following variant of this equation

$$\mathbf{S}_{\mathrm{x;noise}} = \mathbf{A}_x (\mathbf{K}_x^T \mathbf{S}_{\mathrm{y;noise}} \mathbf{K}_x + \mathbf{R}_x)^{-1}. \tag{16}$$

All details mentioned in the context of the mapping of measurement noise on temperature applies also to trace gas retrievals.

### 7.2  Propagation of temperature and pointing errors

Temperature and pointing retrieval errors propagate onto the trace gas retrievals. Temperature and pointing retrieval errors are correlated. These correlations – as well as correlations in the altitude domain – have to be considered for the error estimation of trace species. The error components of $\boldsymbol{TLOS}$ are, along with the respective correlation information, represented by the

covariance matrix $\mathbf{S}_{\mathrm{TLOS;component}}$.

Some components of the temperature and pointing errors contribute to entangled errors. Thus, the respective error components contributing to the temperature and pointing error have to be propagated separately. Some of these components contribute to the random error and others contribute to the systematic error. For this reason, we report them component-wise. The following temperature and pointing error components are considered: measurement noise, gain calibration uncertainty, offset

calibration uncertainty, frequency calibration errors, and uncertainties of spectroscopic data.

### 7.2.1  Propagated temperature and pointing noise

The mapping of noise on temperature and pointing information is characterized by the covariance matrix $\mathbf{S}_{\mathrm{x;TLOS\_random}}$ and is evaluated with Eq. (6) where $\mathbf{S}_{\mathrm{b;meas}}$ is specified as $\mathbf{S}_{\mathrm{TLOS;noise}}$. This error covariance matrix represents the noise component of the retrieved temperatures and tangent altitudes of the limb sequence under evaluation. Here we do not need the full covariance

matrix of the temperature and pointing retrieval, but only those blocks which refer to temperature and tangent altitude information. Entries related to the background continuum and gases fitted jointly with temperature and tangent altitudes play no role here. The entries of $\mathbf{S}_{\mathrm{TLOS;noise}}$ are available as a by-product of the temperature and tangent altitude retrieval. Due to their correlated nature, temperature and pointing/tangent altitude errors have to be propagated jointly rather than separately.

$\mathbf{K}_b$ in Eq. (6) is specified as the Jacobian $\mathbf{K}_{\mathrm{TLOS}}$ representing the sensitivities of the radiance in the analysis windows of

the target gas with respect to changes in temperatures and tangent altitudes. $\mathbf{K}_{\mathrm{TLOS}}$ is not the same as the Jacobian used in the





temperature and pointing retrieval, because it refers to the spectral radiances in the microwindows used for the retrieval of the target gas $x$.

Temperature and pointing noise contribute to the random error of the target gas retrieval.

### 7.2.2 Propagated temperature and tangent altitude errors due to spectral shift

A correction of the supposedly less than perfect frequency calibration of the spectra is performed prior to the retrieval of temperature and tangent altitudes as described in Kiefer et al. (2021). These authors also provide estimates of the response of the retrieved temperatures and tangent altitudes to the estimated residual frequency calibration error (See, Section 6.3.3). Again, the response $\Delta_{\text{TLOS;shift}}x$ of the retrieval of trace gas profile $x$ to the temperature and pointing uncertainty due to the spectral shift uncertainty is estimated by the perturbation approach using Eq. (7), where $F_{\text{perturbed}}$ are the spectral radiances
obtained with a radiative transfer calculation using temperatures and pointing perturbed by $\Delta_{\text{shift}}TLOS$, and where $F_{\text{nominal}}$ are the spectral radiances obtained with the spectral shift used for the retrieval.

$$\Delta_{\text{TLOS;shift}}x = -\mathbf{G}\left(F(x;TLOS + \Delta_{\text{shift}}TLOS) - F(x;TLOS)\right) \tag{17}$$

Since temperature and pointing information in terms of tangent altitudes are jointly retrieved in one inversion step, errors of these quantities are correlated, and perturbations are made in one step. $\Delta_{\text{shift}}TLOS$ vectors are available from Kiefer et al.
(2021), who evaluated this quantity for the representative atmospheric conditions listed in Section 5 by perturbation studies. Since the spectral shift correction is performed for each limb scan separately, all related errors contribute to the random error budget. And since the spectral shift correction provides only one scalar value per limb scan and microwindow, resulting target mixing ratio errors are fully correlated in the altitude domain, i.e.,

$$\text{covar}_{\text{shift-tlos};i,j} = \sigma_{\text{shift-tlos};i}\sigma_{\text{shift-tlos};j}, \tag{18}$$

where $\text{covar}_{\text{shift-tlos};i,j}$ is the covariance between the errors of the target species due to the propagated shift-induced temperature and pointing errors, $\sigma_{\text{shift-tlos}}$ at altitude levels $i$ and $j$.

### 7.2.3 Propagation of temperature and tangent altitude errors due to offset calibration uncertainties

Retrieved temperatures and tangent altitudes are susceptible to offset calibration uncertainties (see Sect 6.3.2). The relevant component of the offset uncertainty that is not removed by joint-fitting the offset along with the target variables is dominated
by noise in the deep space measurements. Related temperature and pointing errors propagate onto the error budget of the target species. Their contribution is estimated using Gaussian error propagation according to Eq. (6), where $\mathbf{K}_b$ is the sensitivity of the radiances used for the retrieval of the target gas to temperatures and tangent altitudes and where $\mathbf{S}_{\text{b;meas}} = \mathbf{S}_{\text{TLOS;offset}}$, i.e., the covariance matrix of temperature and pointing errors due to offset uncertainties. $\mathbf{S}_{\text{TLOS;offset}}$ has relevant off-diagonal entries for the following reasons: (1) Offset measurements use interferograms with a shorter maximum optical path difference than the
scene spectra but are finally zero padded to the length of the scene interferograms (corresponding to a Fourier interpolation in the spectral domain to achieve the same sampling as the scene spectra), (2) apodization is applied, and (3) offset measurements





are used for multiple tangent altitudes. The offset noise variances are calculated from the noise equivalent spectral radiances shown in Fig. 8 of Kleinert et al. (2018). Since the offset uncertainties vary randomly in the wavenumber domain, and since the spectral analysis windows of temperature along with pointing are generally different from those used for the retrieval of the target species, this particular calibration uncertainty does not fall into the category of entangled errors but can be treated as an independent error component. This error component contributes to the random error of the target gas.

### 7.2.4 Propagated temperature and tangent altitude errors due to gain calibration and spectroscopic data uncertainties

Further error components contributing to the temperature and pointing random error are gain calibration uncertainties (Section 6.3.1) and uncertainties in the spectroscopic data used (Section 6.4). On the supposition that spectroscopic data errors are fully correlated in the spectral domain, related propagated temperature and tangent altitude errors fall in the category of entangled errors and are discussed along with the respective direct propagation of $CO_2$ spectroscopic uncertainties onto the target gas retrieval (Section 7.5).

If the target gas is chiefly retrieved in the MIPAS A band, used for the temperature and tangent altitude retrieval, also propagated temperature and tangent altitude errors due to gain calibration belong into the category of entangled errors and are discussed along with the directly propagated gain calibration errors (Section 7.4.1). The situation is different if the target gas is retrieved in another MIPAS band. The dominant gain error components (especially those caused by nonlinearity) are correlated only within MIPAS bands but not between MIPAS bands (Kleinert et al., 2018). This implies that propagated gain calibration errors of temperature and tangent altitudes are uncorrelated with the gain calibration error of the target gas and have thus to be treated as independent error components. In this case, the propagation of the gain-related temperature and tangent altitude error on the target gas concentration is estimated with Eq. (7), where

$$\boldsymbol{F}_{\text{perturbed}} = \boldsymbol{F}(\boldsymbol{x}; \boldsymbol{TLOS} + \Delta_{\text{gain}}\boldsymbol{TLOS}). \tag{19}$$

### 7.3 Uncertainties of interfering species

Uncertainties of interfering species, i.e., species that contribute to the signal in the microwindows of the target gas, have, broadly speaking, small impact on the retrieved mixing ratios of the target species. This is because (a) the microwindows have been defined such that the signal of interfering species is minimized, (b) the sequence of operations is such that the abundances of strong emitters are retrieved first and are thus available when weak emitters are analyzed, (c) for most interfering species retrieved mixing ratios for the actual conditions are available from earlier MIPAS data versions, and (d) in cases of an appreciable influence of the interfering gas on the retrieved profile of the target gas, the interfering gas is jointly fitted with the target gas and thus does not contribute to the parameter error budget but is accounted for already in the noise covariance matrix of the combined target-interferent retrieval. Nevertheless, the propagation of the uncertainties of interfering species is considered. The error estimation schemes used depend on the source of the information on the interfering species. Sources of information on these constituents' abundances and their uncertainties are:





1. Preceding retrievals in the sequential retrieval chain;

2. MIPAS version 5 data;

3. the MIPAS initial guess database.

In the following, error propagation for these cases is discussed. If a certain constituent is a strong interferent, that is to say, causes a large signal in the microwindows of the target gas, occasionally this constituent is fitted jointly with the target gas. In some cases, this approach is chosen even if the abundance is already known from a preceding retrieval step. The reason

behind this approach is to avoid spectral residuals caused by spectroscopic inconsistencies between the microwindows where the interfering constituent has been retrieved and the microwindows where the target gas is retrieved. In this case, the effect of the interferent chiefly is that its consideration in the retrieval slightly increases $\mathbf{S}_{\text{x;noise}}$ and no extra treatment of the interferent is needed in the error budget. In these joint retrievals, the regularization of the interferent is chosen sufficiently weak to ignore any smoothing error crosstalk between the interferent and the target gas.

For interfering gases that were not jointly fitted along with the target gas, the error components are evaluated for each gas separately. The only exception are gases which were jointly retrieved in a preceding retrieval, where therefore inter-gas covariances have to be considered.

### 7.3.1   VMR information from preceding retrievals

MIPAS spectra contain contributions of tens of different species. The simultaneous inversion that provides all these mixing ratio

profiles in one single inversion is not practicable. Instead, the retrieval is decomposed into a series of retrievals, each providing information on typically only one, occasionally a few, species, and each using spectral microwindows which contain the largest possible amount of information on the target gas while contributions by interfering gases are kept small. The retrieval chain is organized in a way that first the mixing ratio profiles of those trace gases are retrieved that make major signal contributions to the spectrum. When the retrieval of the abundances of minor contributors follows later in the retrieval chain, the concentrations

of those gases retrieved earlier in the retrieval chain are already known. Also their noise covariance matrix is available from the preceding retrievals and is used to analyze the error propagation onto the target gas profile, using Eq. (6).

The parameter error covariance matrix $\mathbf{S}_{\text{b;noise}}$ is specified as that block of the resulting covariance matrix from the preceding retrieval that refers to the profile of the interfering gas. In cases when two interfering species were jointly retrieved in the preceding steps, both related blocks and the respective covariance blocks are needed. Other entries of the covariance matrix of

the preceding retrieval need not to be considered here, because the entries of the Jacobian $\mathbf{K}_{\text{b}}$ operating on them would be zero anyway. This Jacobian is specified in this application to represent the sensitivities of the spectral radiances used for the target gas retrieval to the abundancies of the interfering species.

Other uncertainties in the gas concentrations from preceding retrievals (gain error, spectroscopic data uncertainties etc.) belong in the category of entangled errors. The direct errors due to the error source under consideration and the propagated

error due to the impact of the error under consideration on the retrieved abundance of the interfering species have opposite sign which leads to cancellation. Since, due to the way MIPAS data processing is organized, error contributions by interfering





species are generally small, any net effects of these entangled errors that may survive the error cancellation are considered as negligible.

The error components due to uncertainties in the concentrations of interfering species from preceding retrieval steps con-
tribute to the random error of the target gas retrieval, because their systematic components do not effectively propagate due to the compensation mechanism of the entangled errors.

### 7.3.2 VMR information from MIPAS V5

It is not possible to organize the MIPAS retrievals in a way that all interfering gases are known from retrievals performed earlier in the retrieval chain. Some minor interferences from species that are retrieved only later in the retrieval chain do occur
in the microwindows of the target gas. However, earlier MIPAS data versions of these species are often available, e.g., from MIPAS version 5. Also for these MIPAS retrievals, error covariance matrices $\mathbf{S}_{b;noise}$ are available which can be used as $\mathbf{S}_{b;meas}$ in Eq. (6). All said in Section 7.3.1 applies *mutatis mutandis* also to the propagation of uncertainties in the abundancies of interfering species taken from the version 5 MIPAS analysis.

Also these error components due to uncertainties in the concentrations of interfering species from version 5 MIPAS retrievals
contribute chiefly to the random error of the target gas retrieval.

### 7.3.3 VMR information from the MIPAS initial guess database

For interfering gases not yet retrieved from MIPAS spectra, neither version 8 nor V5, mixing ratios and uncertainty estimates from the initial guess database are used. All said on this issue in the context of temperature and tangent altitude error estimation (Section 6.2.2) applies to trace gas error budgets as well.

Assumed uncertainties of $CO_2$ mixing ratios, however, deserve an extra treatment, because they lead to (serious) entangled errors. This is because the retrieval of temperature and tangent altitudes relies on $CO_2$ lines, which causes an entangled error. The entangled nature of this error component has two consequences: First, the perturbed spectra in Eq. (7) have to be calculated as

$$\boldsymbol{F}_{\text{perturbed}} = \boldsymbol{F}(\boldsymbol{x}; \boldsymbol{b}_{\text{CO2}} + \Delta\boldsymbol{b}_{\text{CO2}}, \boldsymbol{TLOS} + \Delta_{CO2}\boldsymbol{TLOS}), \qquad (20)$$

where $\boldsymbol{b}_{\text{CO2}}$ and $\Delta\boldsymbol{b}_{\text{CO2}}$ are the $CO_2$ profiles used and their 1-$\sigma$ perturbations. And second, the propagation of $\Delta_{\text{VMR(CO2)}}\boldsymbol{TLOS}$ implies, that this error component has to be considered also for the error budget of target species whose microwindows do not contain any sizeable $CO_2$ signal. As discussed above, for nominal MIPAS measurements $CO_2$ mixing ratio uncertainties are deemed to contribute to the random error, while for MA/UA measurements a systematic component has to be considered.

### 7.3.4 Smoothing error crosstalk

As already mentioned, in some cases the abundances of interfering species are fitted jointly with those of the target species. This option has been chosen particularly when the abundances of the interfering species pre-retrieved in an earlier step in the retrieval chain do not fit the associated lines in the current target microwindow well. A possible cause for such a behaviour are



inconsistencies in the spectroscopic data in the microwindows where the interferents were retrieved, and the microwindows of the current retrieval step. The purpose of fitting the interferents again is simply to remove related spectral residuals and to

minimize related error propagation. Since the microwindows of the current retrieval step include only little information on the interferents, related results are discarded; they do neither supersede nor complement the results from the earlier retrievals when the interferents were the target species.

Due to the limited amount of information on the interferents, their retrieval has to be heavily regularized in some cases. The regularization chosen is a Tikhonov-type smoothing regularization where a squared first order finite difference operator is

included in the cost function (See, e.g. Kiefer et al., 2021, and references therein). Thus, little information in terms of degrees of freedom is gained. This is tolerable, because the fine structure of the interferents' profiles are available from the earlier retrievals, whose results are used as a priori of the subsequent joint retrieval. That is to say, the fine structure of the interferents' profiles comes from the original retrieval, which is used as a priori, and survives the new joint retrieval, while the information on the total amounts, however at poor vertical resolution, comes from the joint retrieval of the current step.

Critical readers might argue that jointly retrieved species can cause an error component of the target species. This is because the regularization of the jointly fitted interferent will affect also the target species via the off-diagonal blocks of the averaging kernel matrix. We call this error component 'smoothing error crosstalk' (von Clarmann et al., 2020). We argue, however, that in most of our cases the contribution of the smoothing error crosstalk is negligibly small. The reason is this. The availability of the fine structure of the profiles from the original retrieval of the interferents is by far sufficient to avoid any related appreciable

residuals in the spectra, and the total amount lies in the null-space of the Tikhonov regularization matrix block referring to the interferent and thus cannot cause any smoothing error component. In other words, the regularization term in the cost function chosen can only smooth the profiles differences $\hat{x} - x_a$ but cannot push them as a whole towards larger or smaller values.

An exception is the joint retrieval of temperature and nitric oxide (NO) from MIPAS upper atmosphere observations (Funke et al., 2022). In this particular case information on both retrieval variables, temperature and NO, above 105 km is obtained

from the same spectral lines of the NO fundamental band at 5.3 $\mu$m. Further, no original – and better resolved – retrievals are available *a priori* for these variables. The impact of smoothing error crosstalk on the combined temperature and NO retrieval for upper atmospheric observations was extensively investigated by Bermejo-Pantaleón et al. (2011) for MIPAS version V4O retrievals. In particular, these authors showed that the use of inappropriate nighttime NO a priori profiles in this retrieval version led to a pronounced distortion of the retrieved nighttime temperature profiles by up to 50 K in the lower thermosphere. Bermejo-

Pantaleón et al. (2011) therefore recommended to use the full averaging kernel matrices and a priori vectors (covering the full temperature and NO space and all relevant off-diagonal elements) when model results or correlative measurements are made comparable to MIPAS results. A drawback of this recipe, however, is that not always both temperature and NO information are available from model simulations or correlative measurements. And even if they were, such comparisons would be difficult to interpret because resulting differences cannot unequivocally be attributed to individual parameters. To overcome these problems

and to enable comparisons in single parameter spaces (temperature only or NO concentrations only), we report for V8 retrievals crosstalk error estimates that correspond to the mapping of NO a priori uncertainties on the retrieved temperature profile and vice versa. These error estimates are calculated as $(1-A)S_a(1-A)^T$ (Rodgers, 2000) by using a priori covariance matrices $S_a$





manipulated as follows. For the estimation of the smoothing error crosstalk components due to the constraint on the NO profile in the retrieval, the only non-zero entries in $\mathbf{S}_a$ refer to the NO concentration, while temperature variances and covariances

as well as covariances between temperature and NO are set to zero. Conversely, for the estimation of the smoothing error crosstalk due to the temperature constraint, the only non-zero block in $\mathbf{S}_a$ is the one which contains the temperature variances and covariances.

## 7.4 Calibration uncertainties

The same calibration uncertainties discussed in the context of the temperature and tangent altitude retrieval (Section 6.3) are

also relevant to the retrieval of trace gas abundances. These are gain calibration errors, zero offset calibration errors, frequency shift uncertainties and instrument line shape uncertainties.

### 7.4.1 Gain calibration uncertainties

In a similar manner as for the temperature and tangent altitude error estimation, also the propagation of the random and systematic gain calibration uncertainties on the retrieved trace gas abundances are estimated using perturbation studies with

$$\Delta_{\text{gain}}\boldsymbol{x} = -\mathbf{G}(\boldsymbol{F}_{\text{perturbed}} - \boldsymbol{F}_{\text{nominal}}). \tag{21}$$

Gain calibration errors come into play in trace gas retrievals via two different pathways. Firstly, they affect the trace gas retrieval directly. And secondly, they affect the trace gas retrieval via the propagation of the gain error of temperature and tangent altitudes. We have to distinguish between three different cases: The retrieval of the target gas under assessment uses (1) only spectral lines in the MIPAS A band, where TLOS has been retrieved; (2) only spectral lines in MIPAS bands AB to D;

and (3) spectral lines both in the A band and in other bands.

If the target gas is retrieved in the MIPAS A band (Case 1), where also temperature and tangent altitudes are retrieved, gain and gain-induced temperature and tangent altitude errors belong into the category of entangled errors. We take this into account by calculating $\boldsymbol{F}$(perturbed) with temperature and pointing perturbations as resulting from the error estimation of the combined temperature and tangent altitude retrieval, using

$$\boldsymbol{F}_{\text{perturbed}} = \left(1 + \left(\frac{\Delta y}{y}\right)_A\right)\boldsymbol{F}(\boldsymbol{x}; \boldsymbol{TLOS} + \Delta_{\text{gain}}\boldsymbol{TLOS}), \tag{22}$$

$\boldsymbol{TLOS}$ represents the vector representing the temperature profile and the tangent altitudes retrieved in the preceding step and used for the target gas retrieval, and $\Delta_{gain}\boldsymbol{TLOS}$ is the vector containing the responses of the retrieved temperature profile and tangent altitudes to a positive gain perturbation.





For target gases retrieved in any other than the MIPAS A band (Case 2), this entanglement mechanism does not apply. In
this case, the target gas error component due to the gain calibration error is calculated as

$$\boldsymbol{F}_{\text{perturbed}} = (1 + \Delta \boldsymbol{y}/\boldsymbol{y})\boldsymbol{F}(\boldsymbol{x}). \tag{23}$$

This error component and the mapping of the gain-related temperature and tangent altitude error as calculated with the
perturbation as defined in Eq. (19) are treated as independent errors. If the retrieval uses lines from multiple MIPAS bands
AB, B, C, or D, it is adequate to consider both the systematic and the random components of the gain calibration error of
the different bands as independent errors. This is because the random and systematic components of the gain calibration error
are dominated by components that are highly correlated only within a MIPAS band but uncorrelated between the bands. This
implies that for both the systematic and the random component a perturbation calculation is needed for each band involved.

   The situation is more complicated in cases where spectral lines both in the MIPAS A band and one or more of the other
bands are used (Case 3). Both for the systematic and the random part of the gain error the following approach is used: The
perturbed spectrum is calculated using Eq. (22), but the $\Delta \boldsymbol{y}/\boldsymbol{y}$ term is applied only to radiances in the MIPAS A band. The
error component calculated with this perturbation spectrum accounts for the propagated gain-induced temperature and tangent
altitude error as well as the gain error in the A band. Systematic and random components of the error due to gain calibration
uncertainties in the other bands are estimated using Eq. (23) for each band separately.

### 7.4.2   Zero offset calibration uncertainties

The treatment of zero offset calibration uncertainties in the error estimation of trace gas retrievals follows exactly the scheme
presented for temperature and tangent altitude retrievals in Section 7.2.3. Since the relevant components of the zero offset
calibration uncertainty are independent between different spectral regions, and since the microwindows for trace gas retrievals
are different from those used for the temperature and tangent altitude retrievals, these zero-offset related errors do not fall into
the category of entangled errors. This error component contributes to the random error budget.

### 650   7.4.3   Spectral shift uncertainties

Trace gas retrieval errors due to spectral shift errors are estimated by perturbation studies in the same way as for the temperature
and tangent altitude retrieval (Section 6.3.3).

   In this context it should be mentioned that target concentration uncertainties directly caused by spectral shift uncertainties
and target concentration uncertainties due to temperature and pointing errors caused by spectral shift uncertainties do not fall in
the category of entangled errors. This is, because the response of the target concentration to the spectral shift and the response
of the target concentration to shift-induced temperature and pointing errors is erratic rather than systematic.

   Errors in the retrieved quantities due to the spectral shift uncertainty contribute to the random error and they are fully
correlated in the altitude domain.





### 7.4.4 Instrument line shape uncertainties

As discussed in Section 6.3.4, the only relevant instrument line shape parameter to be considered in the error estimation is the modulation efficiency of the interferometer. Its propagation onto the retrieved trace gas abundances is estimated using Eq. (7), where

$$\boldsymbol{F}_{\text{perturbed}} = \boldsymbol{F}(\boldsymbol{x}; e + \Delta e, \boldsymbol{TLOS} + \Delta_{\text{e}}\boldsymbol{TLOS}) \qquad (24)$$

and where

$$\boldsymbol{F}_{\text{nominal}} = \boldsymbol{F}(\boldsymbol{x}; e, \boldsymbol{TLOS}). \qquad (25)$$

$\boldsymbol{x}$ is the retrieved profile of the target gas, at which the perturbations are evaluated. Scalar $e$ is the nominal modulation efficiency and $\Delta e$ its perturbation by 1 $\sigma$. $\Delta_{\text{e}}\boldsymbol{TLOS}$ is the response of the temperature and pointing retrieval to a perturbation of $e$ by $\Delta e$. Since the direct effect of $\Delta e$ and its indirect effect via $\Delta_{\text{e}}\boldsymbol{TLOS}$ are entangled errors, their perturbations are evaluated in one run of the forward model, in order to get the compensation effects correctly. This error component contributes to the 670 systematic error and is fully correlated in the altitude domain.

### 7.5 Uncertainties in spectroscopic data

Uncertainties in line intensities and broadening coefficients are fully correlated in the altitude domain, as far as the same microwindows are used for all altitudes, and they contribute chiefly to the systematic error. The problem of unknown error correlations between different lines of the same gas, that has been discussed in Section 6.4 applies also to trace gas retrievals.

### 7.5.1 Line intensities

The response of the retrieval of a target gas to errors in the target gas' line intensities is estimated by perturbation using Eq. (7), following the scheme discussed for temperature and tangent altitudes in Section 6.4.1. Errors in the intensities of $CO_2$ lines deserve special attention in this context. There is no systematic coupling mechanism between the intensity-induced target gas error and the intensity-induced temperature and pointing error. Therefore, this error component is estimated independently of 680 the error component due to the uncertain line intensities of the target species. For this purpose, also Eq. (7) is used, where

$$\boldsymbol{F}_{\text{perturbed}} = \boldsymbol{F}(\boldsymbol{x}; \boldsymbol{LI}_{\text{CO2}} + \Delta\boldsymbol{LI}_{\text{CO2}}, \boldsymbol{TLOS} + \Delta_{\text{LI(CO2)}}\boldsymbol{TLOS}) \qquad (26)$$

and where

$$\boldsymbol{F}_{\text{nominal}} = \boldsymbol{F}(\boldsymbol{x}; \boldsymbol{LI}_{\text{CO2}}, \boldsymbol{TLOS}). \qquad (27)$$

$\boldsymbol{LI}_{\text{CO2}}$ are the intensities of the $CO_2$ lines affecting the signal in the microwindows of the target gas. $\Delta\boldsymbol{LI}_{\text{CO2}}$ is the vector of 685 intensity perturbations, all with the same sign but individual amount. $\Delta_{\text{LI(CO2)}}\boldsymbol{TLOS}$ is the response of the temperature and pointing retrieval to $CO_2$ line intensity perturbations by 1 sigma. The perturbation is made for the entire $\boldsymbol{TLOS}$ vector in one





step, where the signs of the $\Delta_{\text{LI(CO2)}}TLOS$ components are considered. Due to the entangled nature of the effects of $\Delta LI_{\text{CO2}}$ and $\Delta_{\text{LI(CO2)}}\boldsymbol{TLOS}$, $\boldsymbol{F}_{\text{perturbed}}$ is evaluated for both these effects in one step.

Contrary to temperature and tangent altitude information, errors of pre-retrieved concentrations of interfering gases affect only the signal in the lines of these interfering gases. Here the compensation mechanism discussed in Section 4.4 takes place in full. A too high line intensity of the interfering line will typically cause a too low mixing ratio of the interferent, and the combination of both these error components will produce a signal of the interfering line fairly close to the true signal. Therefore we do not consider the line intensity errors of the interfering species in the error budget of the target species.

### 7.5.2   Broadening coefficients

The propagation of uncertainties of the broadening coefficients onto trace gas mixing ratios follows the scheme described in the previous section for line intensities. Also here the entangled nature of the temperature and tangent altitude errors due to uncertainties of broadening coefficients of $CO_2$ lines has to be taken into account.

     For the evaluation of the propagation of temperature and pointing uncertainties due to $CO_2$ broadening coefficients according to Eq. (7) we use

$$\boldsymbol{F}_{\text{perturbed}} = \boldsymbol{F}(\boldsymbol{x}; \boldsymbol{B}_{\text{CO2}} + \Delta\boldsymbol{B}_{\text{CO2}}, \boldsymbol{TLOS} + \Delta_{\text{B(CO2)}}\boldsymbol{TLOS}) \tag{28}$$

and

$$\boldsymbol{F}_{\text{nominal}} = \boldsymbol{F}(\boldsymbol{x}; \boldsymbol{B}_{\text{CO2}}, \boldsymbol{TLOS}), \tag{29}$$

where $\boldsymbol{B}_{\text{CO2}}$ are the relevant broadening coefficients of the $CO_2$-lines involved, $\Delta\boldsymbol{B}_{\text{CO2}}$ is the respective vector of perturbations, and where $\Delta_{\text{B(CO2)}}\boldsymbol{TLOS}$ is the net response of the temperature and pointing retrieval to perturbations of $CO_2$ broadening 705 coefficients.

     For reasons discussed in the previous section, estimated errors of pre-retrieved interferents due to uncertainties in the broadening coefficients are not considered.

## 8   Further sources of error

In this paper, we concentrate on the assessment of error components that are relevant to temperature, tangent altitudes, and 710 all species retrieved from MIPAS spectra. For the retrieval of products from non-nominal observation modes as well as some gases, observation-mode-specific or gas-specific uncertainties may be relevant, in particular, if non-LTE is considered. The assessment of these uncertainties will be discussed in the corresponding retrieval papers, where relevant. The assessment of these uncertainties will either be based of Eq. (7) or simply on sensitivity studies, where the results of retrievals using different retrieval setups are compared.



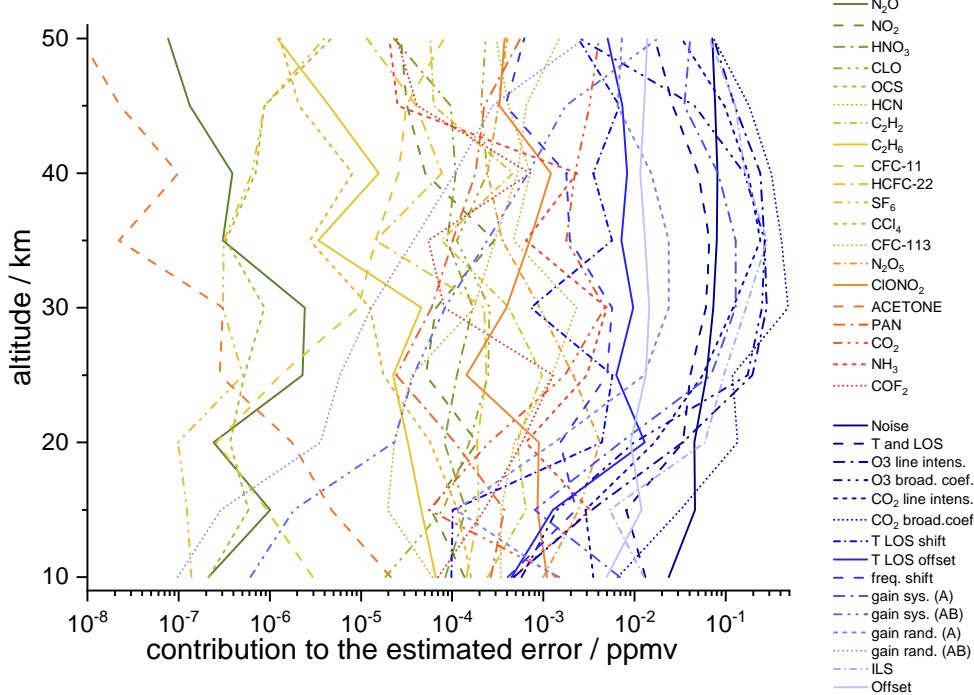

**Figure 1.** The error budget (absolute values) of a MIPAS ozone retrieval from a limb scan recorded at 20.43°S, 18.25°E during Envisat orbit 38517 on 12 July 2009, 21:12 UT.

## 9 A case study: ozone

The error propagation approach presented above is discussed using an ozone retrieval from a MIPAS nighttime measurement at 20.43°S, 18.25°E during Envisat orbit 38517 on 12 July 2009 as a case study. Details of the underlying retrieval procedure are reported by M. Kiefer et al. (2022). Resulting error components for selected altitudes are presented in Fig. 1 and Table B1.

Since a large number of strong ozone lines is available for the retrieval, noise makes only a moderate contribution to the error budget of MIPAS ozone. Instead, the error budget is driven by uncertainties in spectroscopic data, namely line intensities and broadening coefficients. Both spectroscopic data uncertainties of the target gas ozone and those of $CO_2$ are important. The latter affect the ozone retrieval mainly via tangent altitude errors which propagate onto the ozone retrievals. Further, considerable errors are caused by calibration uncertainties, associated both with gain and offset calibration. Calibration uncertainties related to the MIPAS A band have a much larger effect on the ozone retrieval than those related to the AB band, simply because the majority of spectral lines used for the ozone retrieval are situated in the MIPAS A band (M. Kiefer et al., 2022). Also instrument line shape (ILS) and tangent altitude (LOS) uncertainties make considerable contributions to the MIPAS ozone error budget.





Broadly speaking, spectroscopic uncertainties along with instrument and calibration-related as well as uncertainties in temperature and pointing information outweigh uncertainties in the abundances of interfering species by far. None of these uncertainties exceeds the contribution of measurement noise.

## 10   Representative error estimates

Since it is hard to decide a priori which limb scans are representative of a certain typical atmospheric situation, error estimates are calculated for a large number of observations, as discussed in Section 5. The representative errors are estimated differently for errors inferred using Gaussian error propagation (Eqs. 5 or 6) and errors estimated using perturbation studies (Eq. 7).

For errors estimated via Gaussian error propagation, the variances – and if available, also the covariances – are averaged over all limb scans assigned to a given scenario. Since atmospheric state variables retrieved from MIPAS spectra are represented on a fixed altitude grid, independent of the tangent altitudes of the measurement, this step does not involve any interpolation. The result is one mean variance profile per error component and per scenario. For error components where covariance information is available, the respective covariance matrices are averaged. All these error components are regarded as random error components.

Estimates of random error components resulting from perturbation studies are obtained by arithmetical averaging of the responses to the perturbation. To decide if the error component is chiefly additive or chiefly multiplicative, a linear regression is performed using the responses to the perturbation over the mixing ratios. A predominant axis intercept indicates an absolute (i.e. additive) error component, while a pronounced slope indicates a relative (multiplicative) error component. This empirical assessment supersedes the hitherto intuition-based classification of additive versus multiplicative error components.

Due to the headache error problem, the procedure is slightly more complicated for error components that are labelled 'systematic' and evaluated on the basis of perturbation studies, because modulation of the error through the randomly varying atmospheric state may add a random component. When no sizeable latitude or time dependency of these errors is observed, the systematic part of the error component under investigation is the mean over all limb scans associated with the scenario under assessment. The information on the random component of the headache error lies in the scatter around the mean error. When a sizeable dependency of any explaining variable (latitude, time, whatsoever) is found, a parametric model has to be fitted to the individual errors. In this case, the bias then can be estimated for the actual condition with this model, and the information on the random part of the headache error is included in the unexplained variance around this parameterization. The distinction between additive and multiplicative error components is performed on the basis of a linear regression as described above for the random errors. Most of the error sources assessed by perturbation studies result in multiplicative errors; thus the use of relative rather than absolute errors is adequate for disentangling systematic and random parts of the headache error.

Error covariances of the random components of the headache errors in the altitude domain can be calculated in a straight forward way along with the averaging procedure.





## 11 Aggregation of error estimates

Both the total random error and the total systematic error are calculated by adding the respective component errors in terms of
variances. For this purpose, percentage errors are transformed to absolute errors using the respective reference concentration
profile. Headache errors contribute with their random and systematic parts to both categories. The estimated total error variance
is the sum of the random error variance and the systematic error variance. Since the correlation characteristics in different
domains such as time, altitude, among species, etc. can be different for each error component, data users are advised to work
with the detailed error budget rather than the total error estimates.

## 12 Technical realization

A control program runs over all limb scans associated with the scenario under assessment. For each limb scan, this program calls
the radiative transfer model KOPRA for calculation of the reference spectra $F_{nominal}$ and the perturbed spectra $F_{perturbed}$ needed
for calculation of error components according to Eq. (7). Further it provides the gain functions $\mathbf{G}$ and covariance matrices
$\mathbf{S}_{x,noise}$ of the target species and, as far as available, the covariance matrices $\mathbf{S}_{b,noise}$ of parameter errors as well as the required
Jacobians $\mathbf{K}_b$. With this information on the current limb scan available, the error estimator is called. This program extracts the
relevant information from the available covariance matrices $\mathbf{S}_{x,noise}$ and $\mathbf{S}_{b,noise}$, reads the difference spectra obtained from the
perturbed and reference spectra, and estimates the error components of the limb scan under analysis. For each limb scan under
consideration, the error estimator provides the error estimates for each error component, based on Eqs. (5–7).

Based on the individual estimated error components, a post processing routine performs all the statistics over the limb scans
in the scenario under assessment including the disentangling of the headache error as described in Section 10. For the reference
scenarios, the resulting error estimates are reported, component-wise, random and systematic, and are deemed representative of
the scenario they are assigned to. Our data come with an error-class-id for each profile that, based on the information provided
in Table A6, enables the data user to decide for any MIPAS measurement which scenario is applicable. The user who needs
only error budgets for the categories systematic versus random error and absolute versus relative errors has not to refer to
these error-class-ids, because for these categories the related subtotal errors are transformed to and provided for each single
observation.

## 13 Conclusions

This paper presents an overview of error estimation scheme used for temperatures and trace gas concentrations retrieved from
MIPAS spectra with the IMK/IAA data processor. It represents a best effort to make the error reporting compliant with the
recommendations by the TUNER activity as summarized in von Clarmann et al. (2020). Arguably, some error components
are not specified as accurately as one would like to have them. This holds particularly for uncertainties in spectroscopic data,
namely, line intensities and broadening coefficients. For many species, these uncertainties belong to the leading error sources.
The main drawback is, that no information on spectroscopic error correlations between the various lines of a gas is provided.


The related target gas error largely depends on this correlation. Further, it is a truism that unrecognized or unquantified error
sources cannot be considered. Validation studies will show how realistic the estimated random and systematic errors are, how
complete the error budget is, and how justified the ingoing assumptions are.

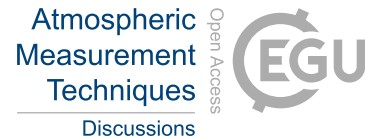

# Appendix A: Sample atmospheres



**Table A1.** Measurements used for error analysis of nominal HR measurements.

| name | date(s) | latitude range(s) | SZA range |
|---|---|---|---|
| | YYYYMM | degrees | degrees |
| Northern polar winter day | [200301,200302] | [65,90] | [0,90] |
| Northern polar winter night | [200301,200302] | [65,90] | [95,180] |
| Northern polar spring day | [200303,200305] | [65,90] | [0,90] |
| Northern polar spring night | [200303,200305] | [65,90] | [95,180] |
| Northern polar summer day | [200306,200308] | [65,90] | [0,90] |
| Northern polar summer night | [200306,200308] | [65,90] | [95,180] |
| Northern polar autumn day | [200309,200311] | [65,90] | [0,90] |
| Northern polar autumn night | [200309,200311] | [65,90] | [95,180] |
| Northern midlatitude winter day | [200301,200302] | [40,60] | [0,90] |
| Northern midlatitude winter night | [200301,200302] | [40,60] | [95,180] |
| Northern midlatitude spring day | [200303,200305] | [40,60] | [0,90] |
| Northern midlatitude spring night | [200303,200305] | [40,60] | [95,180] |
| Northern midlatitude summer day | [200306,200308] | [40,60] | [0,90] |
| Northern midlatitude summer night | [200306,200308] | [40,60] | [95,180] |
| Northern midlatitude autumn day | [200309,200311] | [40,60] | [0,90] |
| Northern midlatitude autumn night | [200309,200311] | [40,60] | [95,180] |
| Tropics day | [200303,200305] | [-20,20] | [0,90] |
| Tropics night | [200303,200305] | [-20,20] | [95,180] |
| Southern midlatitude winter day | [200306,200308] | [-60,-40] | [0,90] |
| Southern midlatitude winter night | [200306,200308] | [-60,-40] | [95,180] |
| Southern midlatitude spring day | [200309,200311] | [-60,-40] | [0,90] |
| Southern midlatitude spring night | [200309,200311] | [-60,-40] | [95,180] |
| Southern midlatitude summer day | [200301,200302] | [-60,-40] | [0,90] |
| Southern midlatitude summer night | [200301,200302] | [-60,-40] | [95,180] |
| Southern midlatitude autumn day | [200303,200305] | [-60,-40] | [0,90] |
| Southern midlatitude autumn night | [200303,200305] | [-60,-40] | [95,180] |
| Southern polar winter day | [200306,200308] | [-90,-65] | [0,90] |
| Southern polar winter night | [200306,200308] | [-90,-65] | [95,180] |
| Southern polar spring day | [200309,200311] | [-90,-65] | [0,90] |
| Southern polar spring night | [200309,200311] | [-90,-65] | [95,180] |
| Southern polar summer day | [200301,200302] | [-90,-65] | [0,90] |
| Southern polar summer night | [200301,200302] | [-90,-65] | [95,180] |
| Southern polar autumn day | [200303,200305] | [-90,-65] | [0,90] |
| Southern polar autumn night | [200303,200305] | [-90,-65] | [95,180] |





**Table A2.** Measurements used for error analysis of RR nominal measurements.

| name | date(s) YYYYMM | latitude range(s) degrees | SZA range degrees |
|---|---|---|---|
| Northern polar winter day | [200901,200902] | [65,90] | [0,90] |
| Northern polar winter night | [200901,200902] | [65,90] | [95,180] |
| Northern polar spring day | [200903,200905] | [65,90] | [0,90] |
| Northern polar spring night | [200903,200905] | [65,90] | [95,180] |
| Northern polar summer day | [200906,200908] | [65,90] | [0,90] |
| Northern polar summer night | [200906,200908] | [65,90] | [95,180] |
| Northern polar autumn day | [200909,200911] | [65,90] | [0,90] |
| Northern polar autumn night | [200909,200911] | [65,90] | [95,180] |
| Northern midlatitude winter day | [200901,200902] | [40,60] | [0,90] |
| Northern midlatitude winter night | [200901,200902] | [40,60] | [95,180] |
| Northern midlatitude spring day | [200903,200905] | [40,60] | [0,90] |
| Northern midlatitude spring night | [200903,200905] | [40,60] | [95,180] |
| Northern midlatitude summer day | [200906,200908] | [40,60] | [0,90] |
| Northern midlatitude summer night | [200906,200908] | [40,60] | [95,180] |
| Northern midlatitude autumn day | [200909,200911] | [40,60] | [0,90] |
| Northern midlatitude autumn night | [200909,200911] | [40,60] | [95,180] |
| Tropics day | [200903,200905] | [-20,20] | [0,90] |
| Tropics night | [200903,200905] | [-20,20] | [95,180] |
| Southern midlatitude winter day | [200906,200908] | [-60,-40] | [0,90] |
| Southern midlatitude winter night | [200906,200908] | [-60,-40] | [95,180] |
| Southern midlatitude spring day | [200909,200911] | [-60,-40] | [0,90] |
| Southern midlatitude spring night | [200909,200911] | [-60,-40] | [95,180] |
| Southern midlatitude summer day | [200901,200902] | [-60,-40] | [0,90] |
| Southern midlatitude summer night | [200901,200902] | [-60,-40] | [95,180] |
| Southern midlatitude autumn day | [200903,200905] | [-60,-40] | [0,90] |
| Southern midlatitude autumn night | [200903,200905] | [-60,-40] | [95,180] |
| Southern polar winter day | [200906,200908] | [-90,-65] | [0,90] |
| Southern polar winter night | [200906,200908] | [-90,-65] | [95,180] |
| Southern polar spring day | [200909,200911] | [-90,-65] | [0,90] |
| Southern polar spring night | [200909,200911] | [-90,-65] | [95,180] |
| Southern polar summer day | [200901,200902] | [-90,-65] | [0,90] |
| Southern polar summer night | [200901,200902] | [-90,-65] | [95,180] |
| Southern polar autumn day | [200903,200905] | [-90,-65] | [0,90] |
| Southern polar autumn night | [200903,200905] | [-90,-65] | [95,180] |





**Table A3.** Measurements used for error analysis of middle atmosphere measurements.

| name | date(s) | latitude range(s) | SZA range |
| --- | --- | --- | --- |
| | YYYYMM | degrees | degrees |
| Northern polar winter day | [200901,200902] | [65,90] | [0,90] |
| Northern polar winter night | [200901,200902] | [65,90] | [98,180] |
| Northern polar spring day | [200903,200905] | [65,90] | [0,90] |
| Northern polar spring night | [200903,200905] | [65,90] | [98,180] |
| Northern polar summer day | [200906,200908] | [65,90] | [0,90] |
| Northern polar summer night | [200906,200908] | [65,90] | [98,180] |
| Northern polar autumn day | [200909,200911] | [65,90] | [0,90] |
| Northern polar autumn night | [200909,200911] | [65,90] | [98,180] |
| Northern midlatitude winter day | [200901,200902] | [40,60] | [0,90] |
| Northern midlatitude winter night | [200901,200902] | [40,60] | [98,180] |
| Northern midlatitude spring day | [200903,200905] | [40,60] | [0,90] |
| Northern midlatitude spring night | [200903,200905] | [40,60] | [98,180] |
| Northern midlatitude summer day | [200906,200908] | [40,60] | [0,90] |
| Northern midlatitude summer night | [200906,200908] | [40,60] | [98,180] |
| Northern midlatitude autumn day | [200909,200911] | [40,60] | [0,90] |
| Northern midlatitude autumn night | [200909,200911] | [40,60] | [98,180] |
| Tropics day | [200903,200905] | [-20,20] | [0,90] |
| Tropics night | [200903,200905] | [-20,20] | [98,180] |
| Southern midlatitude winter day | [200906,200908] | [-60,-40] | [0,90] |
| Southern midlatitude winter night | [200906,200908] | [-60,-40] | [98,180] |
| Southern midlatitude spring day | [200909,200911] | [-60,-40] | [0,90] |
| Southern midlatitude spring night | [200909,200911] | [-60,-40] | [98,180] |
| Southern midlatitude summer day | [200901,200902] | [-60,-40] | [0,90] |
| Southern midlatitude summer night | [200901,200902] | [-60,-40] | [98,180] |
| Southern midlatitude autumn day | [200903,200905] | [-60,-40] | [0,90] |
| Southern midlatitude autumn night | [200903,200905] | [-60,-40] | [98,180] |
| Southern polar winter day | [200906,200908] | [-90,-65] | [0,90] |
| Southern polar winter night | [200906,200908] | [-90,-65] | [98,180] |
| Southern polar spring day | [200909,200911] | [-90,-65] | [0,90] |
| Southern polar spring night | [200909,200911] | [-90,-65] | [98,180] |
| Southern polar summer day | [200901,200902] | [-90,-65] | [0,90] |
| Southern polar summer night | [200901,200902] | [-90,-65] | [98,180] |
| Southern polar autumn day | [200903,200905] | [-90,-65] | [0,90] |
| Southern polar autumn night | [200903,200905] | [-90,-65] | [98,180] |



**Table A4.** Measurements used for error analysis of upper atmosphere measurements, low solar activity.

| name | date(s) YYYYMM | latitude range(s) degrees | SZA range degrees |
|---|---|---|---|
| Northern polar winter day | [200901,200902] | [65,90] | [0,90] |
| Northern polar winter night | [200901,200902] | [65,90] | [100,180] |
| Northern polar spring day | [200903,200905] | [65,90] | [0,90] |
| Northern polar spring night | [200804,200804] | [65,90] | [100,180] |
| | [200903,200905] | | |
| | [201004,201004] | | |
| Northern polar summer day | [200906,200908] | [65,90] | [0,90] |
| Northern polar summer night | [200806,200808] | [65,90] | [98,180] |
| | [200906,200908] | | |
| | [201006,201008] | | |
| Northern polar autumn day | [200909,200911] | [65,90] | [0,90] |
| Northern polar autumn night | [200909,200911] | [65,90] | [100,180] |
| Northern midlatitude winter day | [200901,200902] | [40,60] | [0,90] |
| Northern midlatitude winter night | [200901,200902] | [40,60] | [100,180] |
| Northern midlatitude spring day | [200903,200905] | [40,60] | [0,90] |
| Northern midlatitude spring night | [200903,200905] | [40,60] | [100,180] |
| Northern midlatitude summer day | [200906,200908] | [40,60] | [0,90] |
| Northern midlatitude summer night | [200906,200908] | [40,60] | [100,180] |
| Northern midlatitude autumn day | [200909,200911] | [40,60] | [0,90] |
| Northern midlatitude autumn night | [200909,200911] | [40,60] | [100,180] |
| Tropics day | [200903,200905] | [-20,20] | [0,90] |
| Tropics night | [200903,200905] | [-20,20] | [100,180] |
| Southern midlatitude winter day | [200906,200908] | [-60,-40] | [0,90] |
| Southern midlatitude winter night | [200906,200908] | [-60,-40] | [100,180] |
| Southern midlatitude spring day | [200909,200911] | [-60,-40] | [0,90] |
| Southern midlatitude spring night | [200909,200911] | [-60,-40] | [100,180] |
| Southern midlatitude summer day | [200901,200902] | [-60,-40] | [0,90] |
| Southern midlatitude summer night | [200901,200902] | [-60,-40] | [100,180] |
| Southern midlatitude autumn day | [200903,200905] | [-60,-40] | [0,90] |
| Southern midlatitude autumn night | [200903,200905] | [-60,-40] | [100,180] |
| Southern polar winter day | [200906,200908] | [-90,-65] | [0,90] |
| Southern polar winter night | [200906,200908] | [-90,-65] | [100,180] |
| Southern polar spring day | [200909,200911] | [-90,-65] | [0,90] |
| Southern polar spring night | [200909,200911] | [-90,-65] | [100,180] |
| Southern polar summer day | [200901,200902] | [-90,-65] | [0,90] |
| Southern polar summer night | [200801,200802] | [-90,-65] | [100,180] |
| | [200901,200902] | | |
| | [201001,201002] | | |
| Southern polar autumn day | [200903,200905] | [-90,-65] | [0,90] |
| Southern polar autumn night | [200903,200905] | [-90,-65] | [100,180] |





**Table A5.** Measurements used for error analysis of upper atmosphere measurements during high solar activity

| name | date(s) | latitude range(s) | SZA range |
|---|---|---|---|
| | YYYYMM | degrees | degrees |
| Northern polar winter day | [201201,201202] | [60,90] | [0,90] |
| Northern polar winter night | [201201,201202] | [60,90] | [100,180] |
| Northern polar spring day | [201103,201105] | [60,90] | [0,90] |
| Northern polar spring night | [201103,201105] | [60,90] | [100,180] |
| Northern polar summer day | [201106,201108] | [60,90] | [0,90] |
| Northern polar summer night | [201106,201108] | [60,90] | [98,180] |
| Northern polar autumn day | [201109,201111] | [60,90] | [0,90] |
| Northern polar autumn night | [201109,201111] | [60,90] | [100,180] |
| Northern midlatitude winter day | [201201,201202] | [40,60] | [0,90] |
| Northern midlatitude winter night | [201201,201202] | [40,60] | [100,180] |
| Northern midlatitude spring day | [201103,201105] | [40,60] | [0,90] |
| Northern midlatitude spring night | [201103,201105] | [40,60] | [100,180] |
| Northern midlatitude summer day | [201106,201108] | [40,60] | [0,90] |
| Northern midlatitude summer night | [201106,201108] | [40,60] | [100,180] |
| Northern midlatitude autumn day | [201109,201111] | [40,60] | [0,90] |
| Northern midlatitude autumn night | [201109,201111] | [40,60] | [100,180] |
| Tropics day | [201103,201105] | [-20,20] | [0,90] |
| Tropics night | [201103,201105] | [-20,20] | [100,180] |
| Southern midlatitude winter day | [201106,201108] | [-60,-40] | [0,90] |
| Southern midlatitude winter night | [201106,201108] | [-60,-40] | [100,180] |
| Southern midlatitude spring day | [201109,201111] | [-60,-40] | [0,90] |
| Southern midlatitude spring night | [201109,201111] | [-60,-40] | [100,180] |
| Southern midlatitude summer day | [201201,201202] | [-60,-40] | [0,90] |
| Southern midlatitude summer night | [201201,201202] | [-60,-40] | [100,180] |
| Southern midlatitude autumn day | [201103,201105] | [-60,-40] | [0,90] |
| Southern midlatitude autumn night | [201103,201105] | [-60,-40] | [100,180] |
| Southern polar winter day | [201106,201108] | [-90,-60] | [0,90] |
| Southern polar winter night | [201106,201108] | [-90,-60] | [100,180] |
| Southern polar spring day | [201109,201111] | [-90,-60] | [0,90] |
| Southern polar spring night | [201109,201111] | [-90,-60] | [100,180] |
| Southern polar summer day | [201201,201202] | [-90,-60] | [0,90] |
| Southern polar summer night | [201201,201202] | [-90,-60] | [100,180] |
| Southern polar autumn day | [201103,201105] | [-90,-60] | [0,90] |
| Southern polar autumn night | [201103,201105] | [-90,-60] | [100,180] |





**Table A6.** Attribution of measurements to scenarios

| Scenario | Latitude | SZA | Month |
|---|---|---|---|
| N polar winter day | 60N-90N | 0-95 | DJF |
| N polar winter night | 60N-90N | >95 | DJF |
| N polar spring day | 60N-90N | 0-95 | MAM |
| N polar spring night | 60N-90N | >95 | MAM |
| N polar summer day | 60N-90N | 0-95 | JJA |
| N polar summer night | 60N-90N | >95 | JJA |
| N polar autumn day | 60N-90N | 0-95 | SON |
| N polar autumn night | 60N-90N | >95 | SON |
| N midlat winter day | 30N-60N | 0-95 | DJF |
| N midlat winter night | 30N-60N | >95 | DJF |
| N midlat spring day | 30N-60N | 0-95 | MAM |
| N midlat spring night | 30N-60N | >95 | MAM |
| N midlat summer day | 30N-60N | 0-95 | JJA |
| N midlat summer night | 30N-60N | >95 | JJA |
| N midlat autumn day | 30N-60N | 0-95 | SON |
| N midlat autumn night | 30N-60N | >95 | SON |
| tropics day | 30S-30N | 0-95 | all |
| tropics night | 30S-30N | >95 | all |
| S midlat winter day | 60S-30S | 0-95 | JJA |
| S midlat winter night | 60S-30S | >95 | JJA |
| S midlat spring day | 60S-30S | 0-95 | SON |
| S midlat spring night | 60S-30S | >95 | SON |
| S midlat summer day | 60S-30S | 0-95 | DJF |
| S midlat summer night | 60S-30S | >95 | DJF |
| S midlat autumn day | 60S-30S | 0-95 | MAM |
| S midlat autumn night | 60S-30S | >95 | MAM |
| S polar winter day | 90S-60S | 0-95 | JJA |
| S polar winter night | 90S-60S | >95 | JJA |
| S polar spring day | 90S-60S | 0-95 | SON |
| S polar spring night | 90S-60S | >95 | SON |
| S polar summer day | 90S-60S | 0-95 | DJF |
| S polar summer nigt | 90S-60S | >95 | DJF |
| S polar autumn day | 90S-60S | 0-95 | MAM |
| S polar autumn night | 90S-60S | >95 | MAM |



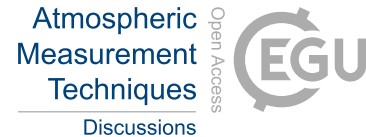

**Appendix B: Case Study**



**Table B1.** Estimated errors of an ozone retrieval .

| Error source | Contribution to the estimated error/ppmv | | | | | | | | |
|---|---|---|---|---|---|---|---|---|---|
| | Altitude/km | | | | | | | | |
| | 10. | 15. | 20. | 25. | 30. | 35. | 40. | 45. | 50. |
| Noise | 2.38e-02 | 4.62e-02 | 4.54e-02 | 6.15e-02 | 7.31e-02 | 7.95e-02 | 8.10e-02 | 7.68e-02 | 7.11e-02 |
| T and LOS | 1.32e-02 | 8.10e-03 | 3.41e-02 | 4.09e-02 | 6.06e-02 | 6.60e-02 | 5.01e-02 | 2.63e-02 | 1.68e-02 |
| $N_2O$ | 2.13e-07 | 1.01e-06 | 2.42e-07 | 2.29e-06 | 2.41e-06 | 3.05e-07 | 3.90e-07 | 1.33e-07 | 7.69e-08 |
| $NO_2$ | 1.37e-04 | 5.91e-05 | 1.88e-04 | 5.14e-05 | 6.66e-05 | 2.89e-04 | 5.04e-05 | 3.19e-05 | 2.73e-05 |
| $HNO_3$ | 8.42e-05 | 1.41e-04 | 8.19e-05 | 1.13e-04 | 1.69e-04 | 8.81e-05 | 1.09e-04 | 9.66e-05 | 2.31e-05 |
| CLO | 1.86e-05 | 7.98e-05 | 1.54e-04 | 2.49e-04 | 2.56e-04 | 1.13e-04 | 2.42e-04 | 2.08e-04 | 2.33e-04 |
| OCS | 2.14e-07 | 5.85e-07 | 3.69e-07 | 5.27e-07 | 8.76e-07 | 3.31e-07 | 6.99e-07 | 8.76e-07 | 4.73e-06 |
| HCN | 3.41e-04 | 3.43e-04 | 4.84e-04 | 1.24e-03 | 2.31e-03 | 4.72e-04 | 7.16e-04 | 4.14e-04 | 3.01e-04 |
| $C_2H_2$ | 2.45e-04 | 6.41e-04 | 5.16e-04 | 9.64e-04 | 4.21e-04 | 7.66e-05 | 4.63e-04 | 6.27e-05 | 5.77e-05 |
| $C_2H_6$ | 6.58e-05 | 4.60e-05 | 3.28e-05 | 2.25e-05 | 4.57e-05 | 3.40e-06 | 1.57e-05 | 4.49e-06 | 1.22e-06 |
| CFC-11 | 2.97e-06 | 9.25e-07 | 2.60e-07 | 1.31e-06 | 9.34e-06 | 1.44e-05 | 2.58e-05 | 3.15e-05 | 8.17e-05 |
| CFC-22 | 1.44e-04 | 1.96e-04 | 1.88e-04 | 2.37e-04 | 2.26e-04 | 1.45e-05 | 7.72e-05 | 4.03e-05 | 1.16e-05 |
| $SF_6$ | 1.38e-07 | 1.24e-07 | 9.89e-08 | 5.10e-07 | 3.05e-07 | 3.12e-07 | 6.40e-07 | 8.74e-07 | 3.96e-06 |
| $CCL_4$ | 1.61e-04 | 1.14e-04 | 5.75e-05 | 1.77e-05 | 1.26e-05 | 2.75e-06 | 8.04e-06 | 2.10e-06 | 1.29e-06 |
| CFC-113 | 6.09e-05 | 1.97e-05 | 2.33e-05 | 3.63e-05 | 1.74e-04 | 2.94e-04 | 5.21e-04 | 6.43e-04 | 1.49e-03 |
| $N_2O_5$ | 1.00e-03 | 2.42e-03 | 4.18e-03 | 2.11e-03 | 9.15e-04 | 3.54e-04 | 2.44e-04 | 3.19e-04 | 4.02e-04 |
| $CLONO_2$ | 1.10e-03 | 8.66e-04 | 8.92e-04 | 1.44e-04 | 3.94e-04 | 7.13e-04 | 1.21e-03 | 3.30e-04 | 3.74e-04 |
| ACETONE | 2.13e-05 | 4.72e-06 | 1.75e-06 | 2.81e-07 | 3.06e-07 | 2.22e-08 | 9.93e-08 | 2.29e-08 | 8.41e-09 |
| PAN | 2.65e-04 | 3.71e-04 | 1.07e-04 | 2.45e-05 | 5.93e-05 | 1.01e-04 | 1.80e-04 | 2.25e-04 | 5.63e-04 |
| $CO_2$ | 1.49e-03 | -5.59e-05 | -2.54e-04 | 1.73e-03 | 4.72e-03 | 1.78e-03 | 2.22e-03 | 3.19e-03 | 4.09e-03 |
| $NH_3$ | -4.67e-04 | -3.25e-03 | -6.39e-04 | 2.99e-03 | -4.95e-03 | -6.47e-04 | -2.33e-03 | -2.52e-05 | -2.04e-05 |
| $COF_2$ | 7.01e-05 | -2.26e-04 | 5.87e-04 | 1.28e-03 | -8.20e-05 | 5.48e-05 | 7.16e-04 | 4.12e-05 | -2.46e-05 |
| $CH_3OH$ | -1.01e-24 | 2.94e-25 | 2.55e-25 | 3.35e-24 | 4.94e-23 | 7.84e-23 | 1.38e-22 | 1.61e-22 | -3.08e-22 |
| line intens. | -4.75e-04 | -3.37e-03 | -3.35e-02 | -1.95e-01 | -2.81e-01 | -2.67e-01 | -2.40e-01 | -1.25e-01 | -7.37e-02 |
| broad. coef. | -5.75e-04 | 1.41e-03 | -9.93e-03 | -1.74e-01 | -2.50e-01 | -2.68e-01 | -1.60e-01 | -4.53e-02 | -2.51e-03 |
| $CO_2$ line intens. | 3.51e-03 | -2.89e-03 | -1.66e-02 | 5.58e-02 | 1.21e-01 | 2.39e-01 | 1.76e-01 | 9.68e-02 | 3.29e-02 |
| $CO_2$ broad. coef. | -6.59e-03 | -3.18e-02 | -1.35e-01 | 1.19e-01 | 4.83e-01 | 4.10e-01 | 3.15e-01 | 1.80e-01 | 7.36e-02 |
| T LOS shift | -9.81e-05 | -1.01e-04 | -4.38e-03 | -5.70e-03 | -7.31e-04 | 5.69e-03 | -3.52e-03 | -6.83e-03 | -2.40e-03 |
| T LOS offset | -4.48e-04 | -1.27e-03 | -1.29e-02 | -6.31e-03 | -9.66e-03 | -7.16e-03 | -8.27e-03 | -7.38e-03 | 5.07e-03 |
| freq. shift | 4.06e-04 | -2.28e-03 | -1.53e-03 | -4.98e-03 | -5.68e-03 | -1.95e-03 | -1.79e-03 | -3.85e-04 | -6.18e-04 |
| gain calib. sys. (A) | 6.96e-03 | -8.02e-04 | -7.08e-03 | -6.91e-02 | -1.29e-01 | -1.29e-01 | -8.21e-02 | -3.49e-02 | -4.09e-02 |
| gain calib. sys. (AB) | -6.12e-07 | 1.84e-06 | 2.29e-05 | 3.74e-05 | -8.34e-05 | -2.42e-04 | -6.85e-04 | -1.76e-03 | 1.80e-02 |
| gain calib. rand. (A) | 1.38e-03 | 1.11e-04 | -1.32e-03 | -1.27e-02 | -2.37e-02 | -2.37e-02 | -1.52e-02 | -6.52e-03 | -7.31e-03 |
| gain calib. rand. (AB) | -9.78e-08 | 2.90e-07 | 3.59e-06 | 5.88e-06 | -1.30e-05 | -3.78e-05 | -1.07e-04 | -2.77e-04 | 2.83e-03 |
| ILS | -1.27e-02 | 5.31e-03 | 5.90e-02 | -9.55e-02 | -1.66e-01 | -2.83e-01 | -1.63e-01 | -1.28e-01 | -6.61e-02 |
| Offset | 4.95e-03 | 1.19e-02 | 9.07e-03 | 1.33e-02 | 1.45e-02 | 1.31e-02 | 1.16e-02 | 1.27e-02 | 1.38e-02 |



*Author contributions.* TvC suggested the initial concept and wrote major parts of the paper. NG provided the error estimation control software. UG provided the error estimation software. MK provided the post-processing software. NG provided the sample calculation. MK and BF selected the representative cases. AK provided level-1 expertise. All authors participated in the development of the concept from the initial suggestion to the final version and provided text and comments.

*Competing interests.* TVC and GPS are associate editors of AMT but are not involved in the evaluation of this paper.

*Acknowledgements.* The World Meteorological Organization (WMO) provided travel support through the Stratosphere-troposphere Processes And their Role in Climate (SPARC) project who have selected the TUNER project as a SPARC activity. The International Space Sciene Institute (ISSI) has funded two International Team meetings in Berne at their venue. ESA provided MIPAS level-1 spectra.





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
