# Peer review of "TUNER-compliant error estimation for MIPAS: Methodology"

_Atmospheric Measurement Techniques, 2022_

## Author Comment (AC1)

**Author Reply**

The authors thank Dr Chris Boone for his constructive, insightful and encouraging comments. In our reply, the original comments are printed in **bold face**, our replies are printed in *italic face*, and the resulting modifications in the paper are printed in normal face.

**Comment: This paper offers a detailed accounting of error estimates for retrievals from the MIPAS instrument. The analysis is rigorous and is as complete as can be expected. There will always be error sources that cannot be estimated because there is insufficient information in the measurement system, such as the impact of measuring from a moving platform. Retrievals always implicitly assume a vertical column, but measurements are smeared geographically, which could cause problems if measuring something with high variability or moving across the polar vortex boundary partway through an altitude scan. Note that a changing scene over the course of a single FTS measurement (e.g., when moving through a region of high variability in H2O) would give rise to a contribution to the imaginary component in the Fourier transform, which is the source of the noise information in the analysis, but perhaps the effect would be negligible compared to the noise level for an instrument measuring in emission.**

**Reply:** *We agree that there will always be error sources that cannot be estimated because there is insufficient information. As soon as there is information available to quantify a new error source, this can be done by perturbation or sensitivity studies as mentioned in our Section 8 "Further sources of error". With respect to line shapes, it should be kept in mind that the spectral line shapes in MIPAS spectra are dominated by the ILS and the spectral resolution of MIPAS, so that the true lineshapes are not the leading part in many cases.*
*Horizontal variability along the line of sight and/or along the flight track is indeed an issue. With regard to level-2 related issues, we report the horizontal information smearing and information displacement in our gas-specific retrieval papers, following the method by von Clarmann et al., Atmos. Meas, Tech. 2, 47-54, 2009. We, however, do not think that this issue belongs into the paper under discussion.*
*Beyond this, there is indeed a level-1 related issue due to signal variations during the recording of an interferogram. We agree that this affects the imaginary component of the interferogram that is the basis of our noise estimate. Clouds cause an even larger effect of this type than water vapour variations. In the early phase of MIPAS level-1 processing this caused indeed unrealistic noise estimates. High-pass filtering of the imaginary part of the interferograms (as mentioned in our manuscript) was found to substantially reduce this problem. Noise estimates as provided by ESA along with the latest version of level-1 data are considered as fairly realistic.*

**Comment: There will be systematic errors from using Voigt profiles in the calculated spectrum rather than more accurate line shapes, but one would hope the available uncertainties on the Voigt parameters would encompass this effect.**

**Reply:** *We agree that deviations from the Voigt line shape and the true line shape are hopefully covered by the Voigt parameter uncertainties. Furthermore, details of the line shape are deemed less visible at MIPAS spectral resolution than with better resolving instruments such as ACE-FTS. If need be, related errors can still be evaluated as discussed in Section 8 "Further Sources of Error".*

**Action:** We have added to Section 8: [The assessment of these uncertainties will be discussed in the corresponding retrieval papers, where relevant.] The same holds for error sources not discussed so far, such as inaccurate line shape models. The relevance of such effects is deemed highly dependent on the target gas under analysis. [The assessment of these uncertainties will either be ...]

**Comment: Some of the labels are a bit whimsical (e.g., headache errors), but their meanings are clear. I had to look up some of the Latin phrases, not being familiar with the language.**

**Reply:** *We have tried to find equivalent English expressions, although these sometimes seem to sound a bit clumsy.*

**Action:** *a fortiori*: replaced with "with even greater force" and elsewhere with "with even greater reason"
*mutatis mutandis*: replaced with "with the necessary changes in place"

**Comment: I have no suggestions for changes, other than a few typos and minor changes, listed below**
**Line 51: variable definitions: Every variable is defined except for $\vec{y}$ (unless you count it as being defined by the phrase "the signal y" on line 94, well after the fact.**

**Reply:** *agreed.*

**Action:** We have added the definition to the tabular list of definitions.

**Comment: Line 67: "... denotes the errors source"**
errors → error.
**Comment: Line 325: "witht"**
with.
**Comment: Line 358: " ... between to independent measurement systems"**
to → two.

**Reply:** *Thanks for spotting*

**Action:** all three corrected

**Comment: Line 378: "The component of the instrument line shape error related to the phase does not need to be considered explicitly, because it affects the frequency shift only and thus is implicitly included in $\Delta_{shift}x$."**
**A non-zero phase in the modulation function would imply a physical asymmetry in the ILS (unless it is just a straight line as a function of optical path difference), the effect of which is not just a frequency shift; it affects the shape of the calculated line..**

**Reply:** *We agree that our original wording was incorrect. We have replaced it by a weaker statement.*

**Action:** We have replaced the sentence by: "Spectral shift errors caused by instrument line shape errors do not need to be considered as part of the instrument line shape error, because the total spectral shift is empirically corrected as the first step of the data processing chain (see Fig 1) and the residual spectral shift uncertainty is propagated as an error source in its own right (see Section 6.3.4)"

**Comment: "Table B1: CFC-22"**
**CFC-12**

**Reply:** *Thanks for spotting.*

**Action:** Corrected.

---

## Author Comment (AC2)

**Author Reply to Reviewer #2**

The authors thank reviewer #2 for their constructive, insightful and encouraging comments. In our reply, the original comments are printed in **bold face**, our replies are printed in *italic face*, and the resulting modifications in the paper are printed in normal face.

**Comment: The authors present linear error estimation methodology for the multi-step MIPAS temperature and trace gas retrieval from limb spectra. The approach considers multiple error sources at multiple steps of the retrieval process and provides their mathematical definitions using conventions recently proposed in the collaborative TUNER activity in the remote sensing retrieval community.**
**The technical development of the error estimation methodology is comprehensive and appropriate for MIPAS retrieval system. Multiple elements of this work, which aims to align with the TUNER enterprise, provide welcome exposition and appropriate mathematical detail for the error analysis of atmospheric remote sensing retrievals. In particular, the authors emphasize the dual role of certain unknowns, such as temperature, as fixed parameters or retrieved quantity of interest, depending on the retrieval step in question. The TUNER conventions for error source notation provide additional clarity in the development. Finally, the detailed specification of perturbation-based calculations versus Gaussian propagation is appreciated..**

**Reply:** *The authors thank the reviewer for this encouraging evaluation.*

**Comment: My first major comment is on the balance of methodology versus results in the paper. As currently presented, the paper is very heavy on the methodology development and light on results. After presentation of the only figure in the paper, the last few sections rapidly discuss multiple additional analyses and summaries that have been or could be undertaken without any illustration of their impact or interpretation. It reads very much like an algorithm theoretical basis document (ATBD) as opposed to a presentation of research results or a data users guide. This current format is potentially appropriate for this journal and certain segments of its audience, but broader appeal might be achieved with additional focus on results.**

**Reply:** *The methodological focus of the paper is intentional. This manuscript is planned as a contribution to the AMT MIPAS Special Issue, which will include gas-specific retrieval and validation papers of MIPAS products. The rationale behind the manuscript under review is to summarize those methodological issues that are overarching over specific species retrievals. The idea is to avoid the various species-specific retrieval papers of the AMT MIPAS Special Issue being*

*overloaded with excessive and repetitive technical and methodological detail. All discussion of impact or interpretation of the resulting error budgets depends on the species under analysis and shall thus go mainly into the gas-specific retrieval papers. However, we have identified some results that are of too technical relevance for the ozone retrieval paper by Kiefer et al. These may help to illustrate the concepts described in Sections 10 and 11.*

*Since some of the concepts presented in the paper under review might also be interesting for missions beyond MIPAS, we think it is appropriate to publish these in a journal paper, since ATBD's are typically read only by scientists involved in the respective mission.*

**Action:** In order to avoid to raise wrong expectations of the reader, we have changed the title of the paper to "TUNER-compliant error estimation for MIPAS: Methodology". Beyond this, material about ozone representative errors and error aggregation will be included in Sections 10 and 11.

**Comment: I recommend the authors assess this balance of the presentation in terms of their intended audience. I can imagine the MIPAS data user community would benefit from further presentation of the error estimates and the availability of them as supplementary data. If this is not within the intended scope of the paper, at least it would be useful to discuss how these materials would be made available. The abstract mentions estimated uncertainties available for multiple atmospheric conditions. While the conditions are discussed in different sections, the error estimates are not provided. There is not an explicit presentation of the systematic versus random error estimates.**

**Reply:** *All uncertainty information is species-dependent and will be published in the gas-specific retrieval papers along with the results. Data that are too big to go into a journal paper will be published via the KITopen data repository (with a doi) and linked to the gas-specific papers. We think that the data users will first consult these gas-specific retrieval papers, where they find the detailed error budgets and links to the related data product including the individual error estimates. Those readers who want to know how the error estimates come about, will likely follow the link to the paper under review, describing the error methodology. The data user who does not care about this will not be buried in technical detail when reading the gas-specific papers.*

*The relevance of the individual error components varies from species to species. By the same token, also the fractional contribution of an error source to the systematic and random component is deemed species dependent. These results will thus be reported in the gas-specific retrieval papers and will be made publicly accessible together with the gas abundances for each individual profile and separated in random and systematic error components. It is our intention to avoid redundancies between the papers in the MIPAS Special Issue.*

**Action:** We have added references of the gas-specific papers. In particular, we

have added at the bottom of Section 10: "[... can be calculated in a straight forward way along with the averaging procedure.] These representative error estimates are reported for the particular species under investigation along with the publication of the data product, such as Kiefer et al. (2022) for ozone."
Further, we have added at the bottom of Section 11: "... provided for each single observation (See, e.g., Kiefer at al, 2022 and references therein). In the conclusion, we have inserted: "In this paper we limit ourselves to the methodology as far as it is overarching over the different data products of MIPAS, to support gas-specific analyses as performed, e.g., by Kiefer et al. (2022) for ozone. Furthermore, Information on expected error correlations in various domains (across limb scans, altitudes, gases) has been added."

**Comment: A second major comment is that the early sections of the paper could be aided by some sort of stage-setting schematic or table on the MIPAS retrieval steps and the potential error sources considered at the various steps. While the specific details of the retrieval system are documented elsewhere, a high-level summary of the steps of this complex retrieval system would aid understanding.**

**Reply:** *We like this idea! This is indeed overarching over the individual species and fits well in with this paper.*

**Action:** We have added a data flow diagram that illustrates the use of mixing ratio and other information during various steps of the processing chain.

**Comment: Some specific comments by section are provided below.**
**2. Definition and notation**
**The specification of the notation is useful at this early stage of the paper. It might be worth mentioning that the regularization matrix R translates to the inverse of the a priori covariance matrix in the optimal estimation nomenclature of Rodgers (2000).**

**Reply:** Agreed.

**Action:** We have inserted immediately after the table with the definitions: "If an inverse a priori covariance matrix, $\mathbf{S}_a^{-1}$ is chosen for the regularization matrix $\mathbf{R}$, then this formalism represents the optimal estimation or maximum a posteriori retrieval scheme endorsed by Rodgers(2000). However, other choices are possible (see, e.g., Steck and von Clarmann, 2001).

**Comment: 3. Error propagation**
**Is there any practical difference between the meas and noise error sources in this work or more generally?**

**Reply:** *Yes, there is a difference, at least in the terminology we use. Noise includes only scene noise (accounting for the fact that the emission of photons is a*

*random process) and detector noise. We use the attribute 'measurement' for all errors or uncertainties due to noise and any uncertainty in the characterization of the measurement. E.g., all calibration uncertainties fall into this category.*

**Action:** We have added at the end of the second paragraph of section "Error propagation": "[In this formulation $\mathbf{S}_{y;\mathrm{meas}}$ is a placeholder and will be replaced by the specific type of measurement error under assessment], e.g., noise, calibration errors, etc."

**Comment: The final sentence of the section attempts to convey multiple ideas and information and comes across as a bit confusing. It might be better to have two separate sentences that first identify the constituents retrieved in the logarithmic domain including any special cases. The second can simply mention that the error estimates are similarly computed in the logarithmic domain.**

**Reply:** *Agreed*

**Action:** We have chopped this lengthy sentence into pieces. This paragraph now reads: "The mixing ratios of some species are retrieved in the logarithmic domain. These are $H_2O$, $CO$, $NO$, $NO_2$, and in middle atmosphere (MA), upper atmosphere (UA), and noctilucent cloud (NLC) measurement modes also $O_3$. For specific MA research products also $CH_4$ and $N_2O$ are retrieved in the logarithmic domain. In these cases also the error estimates are performed in the logarithmic domain and finally mapped into the mixing ratio domain."

**Comment: 4. Terminology**
**Regarding the synonymous use of uncertainties and estimated errors, I would agree that this tends to be the case in the remote sensing literature, particularly in the realm of linear analysis of the type in this paper. However, in other disciplines (statistics, uncertainty quantification) with a probabilistic focus on estimation, error is often considered a random variable, a realization of the difference between an estimated quantity of interest and its true value. Uncertainty is a description/summary of a distribution (e.g. standard deviation), which may be a distribution of errors.**

**Reply:** *We intentionally write 'estimated errors', not only 'errors'. with this, we think, we are not in conflict with a terminology where the term 'error' is a random variable that is a realization of the difference between an estimated quantity of interest and its true value.*

**Action:** We have checked the manuscript to make sure that the term 'error' without further qualification occurs only in situations where it can be understood as a random variable denoting a realization of the difference between an estimated quantity of interest and its true value. At some occasions throughout the manuscript, where the term 'error' occured outside of an established composite term, we have included the attribute 'estimated' to avoid any misunderstanding.

**Comment: In the final remarks on headache errors, the authors mention that these will be catalogued appropriately as systematic or random errors. Would it be expected that part of the "headache" induced is that the resulting systematic and random errors from these sources would vary in magnitude across soundings and geophysical conditions?**

**Reply:** *Not quite. We have taken care to evaluate our uncertainties for fairly homogeneous geophysical conditions. With this, large variations of the magnitude should be excluded. But indeed the headache is caused by random variations of the magnitude across soundings within the fairly homogeneous sample.*

**Action:** We have added at the end of this section: "The random component contains the variability due to the respective error source across soundings within a reference scenario, while the systematic component contains the bias."

**Comment: 5. Selected reference scenarios**
**This section seems to suggest that error estimates are available for all of the reference classes. Subsequent sections discuss how the estimates are computed, but any mention of archiving the estimates themselves (e.g. in a collection of data files) is absent. The paper and potential user community would benefit from making this connection between the methodology and availability of products/results.**

**Reply:** *As said earlier, all product-specific results will be made available in the context of the gas-specific publications. These will contain links to publications in the KITopen data repository. These KITopen publications will include the retrieved distributions and the related error estimates on a profile-by-profile basis. In this step each profile is assigned to a specific latitudinal/seasonal/illuminational error class, and the respective errors are scaled to match the volume mixing ratios of the profile in case of percentage errors. The manuscript under review is meant as a presentation and discussion of the methodology, over-arching over the species.*

**Action:** References to the paper (in preparation) by Kiefer et al have been added, which links to the ozone distributions and error estimates.

**Comment: 6. Temperature and pointing retrieval**
**In the first line of section 6.1, is the retrieval really for a "target gas" in this step?**

**Reply:** *Thanks for spotting! This paragraph was initially written for gases; then*

*the manuscript was restructured, and during this operation this silly copy/paste error happened.*

**Action:** Corrected: "The noise error covariance matrix $\mathbf{S}_{\mathrm{TLOS;noise}}$ of the combined temperature and pointing information vector $\vec{TLOS}$ is calculated ..."

**Comment: In Eq. 8, an inverse is needed with the Sy;noise. The same is true for Eq. 16 in section 7**

**Reply:** *yes, indeed. Thanks for spotting!*

**Action:** corrected in both places.

**Comment: What spectroscopic database(s) is/are used in the MI­PAS retrievals? In this section and others, the text sometimes reads "1sigma" and uses the Greek letter in other places (e.g. 6.3.3. versus 6.3.4). This should be checked for consistency.**

**Reply:** *We mostly use HITRAN, but not always the same version; in some cases the most recent version seemed to be inferior compared to preceding versions (missing lines etc). Beyond this, we sometimes use a custom-tailored MI­PAS database, or sometimes data we got directly from the authors. The choice of the spectroscopic data is heavily species-dependent. Thus, this issue will be discussed species by species in the relevant papers, with references to the data actually used.*
*We agree that the sigma notation should be consistent; indeed, the word 'sigma' and the symbol $\sigma$ mean the same thing.*

**Action** We now use $1\sigma$ throughout when sigma is used as an adjective (e.g., $1\sigma$ perturbation) and $1\ \sigma$ when sigma is used as a noun (e.g., perturbation by $1\ \sigma$. Supposedly the copy editors will change this to the publisher's convention.

**Comment: 7. Trace constituents retrieval**
**The remarks on separate versus joint propagation of temperature and pointing errors in 7.2 (line 426) and 7.2.1 (line 438) seem to contra­dict each other. I suspect there is a subtle distinction between the error sources in these two paragraphs, but that should be clarified.**

**Reply:** *Error propagation is always calculated jointly for temperature and point­ing, but separately for the various sources of the TLOS errors. We see that 'separately' may be misunderstood to refer to 'temperature' and 'pointing'.*

**Action:** Changed to: "[...have to be propagated separately] for the different sources of $TLOS$ errors."

**Comment: Line 602: does the bold 1 symbol represent the identity matrix? Also note that Sa is the a priori covariance if not previously defined.**

**Reply:** *yes,the bold 1 symbol was used to represent the identity matrix. The implicit definition of $\mathbf{S}_a$ follows immediately in the text: "These error estimates are calculated as $(\mathbf{I}-\mathbf{A})\mathbf{S}_a(\mathbf{I}-\mathbf{A})^T$ (Rodgers, 2000) by using a priori covariance matrices $\mathbf{S}_a$ manipulated as follows."*

**Action:** We have changed $\mathbf{1}$ to $\mathbf{I}$ because this seems more conventional to us. Further, we have inserted immediately after the equation: "where $\mathbf{I}$ is the identity matrix of the respective dimension"

**Comment: 9. Case study**
**Are there further details available on the Kiefer paper in preparation?**

**Reply:** *Yes, there are: Michael Kiefer, Thomas von Clarmann, Bernd Funke, Maya García-Comas, Norbert Glatthor, Udo Grabowski, Michael Höpfner, Sylvia Kellmann, Alexandra Laeng, Andrea Linden, Manuel López-Puertas, and Gabriele P. Stiller, Version 8 IMK/IAA MIPAS ozone profiles: nominal observation mode. This manuscript is almost ready for submission, and we expect that the publisher will put the manuscript under discussion on hold if it happens to be accepted before the preprint of Kiefer et al. is available (with doi).*

**Action:** The bibliography has been updated.

**Comment: 10. Representative error estimates**
**This section describes a number of post-processing steps that could provide additional insight to the error estimation procedure, but no further results are presented. Some illustration or example of these procedures would be useful here.**

**Reply:** *Although gas-specific information should go into the gas-specific papers, we agree.*

**Action:** We plan to include an example based on the ozone retrieval.

**Comment: 11. Aggregation**
**More information on the decomposition of random and systematic errors would be welcome here. Without them, this section does not seem to add much to the paper.**

**Reply:** *We agree.*

**Action:** Also here we plan to illustrate the content with an example based on ozone.

**Comment: 13. Conclusion**
**The conclusion could use further discussion about what informa-tion/resources are available to data users resulting from this work (e.g. software, supplemental data products, if any)**

**Reply:** *we have prepared publications in the KITopen data repository (with doi), including the data presented here and the error estimation software. These will be made availble prior to publication of the final version of the paper. For each trace gas concentration profile, an error profile based on the methodology and choices explained in this paper here will be made publicly available in the same KITopen publication as the concentration data. For some reason, the macro-command "codedataavailability" seemed not to work in the preprint format but it usually works in the journal format. This will add an endnote to the paper with links to the code and data publications. In AMT this type of information is not usually provided in the Conclusion section.*

**Action:** the entry under "codeanddataavailability" has been prepared and will be activated when the manuscript will be type-set with the journal (as opposed to preprint) style file."

**Comment: Further context would also be useful in the conclusion. Validation studies are mentioned, but do any of these currently ex-ist? Has there been previous error assessment work presented for previous retrieval product versions?**

**Reply:** *Validation studies on version 8 data are under way but unfortunately nothing has been published yet. Previous MIPAS data versions came along with quite detailed error analyses, however not following the methodology presented here. These error estimates are all species-dependent and thus these are better discussed in the context of the species-specific retrieval papers. Instead, we put the focus on the relevance of the separate reporting of systematic versus random errors.*

**Action:** References to Laeng et al. (2014), Plieninger et al. (2016) and Eckert et al. (2016) have been included, where random and systematic error estimates are validated by comparison to scatter of differences and bias between MIPAS and comparison instruments.